# Wnt and Src signals converge on YAP-TEAD to drive intestinal regeneration

Oriane Guillermin[1,2] , Nikolaos Angelis[2], Clara M Sidor[1], Rachel Ridgway[3], Anna Baulies[2], Anna Kucharska[2], Pedro Antas[2], Melissa R Rose[2], Julia Cordero[4], Owen Sansom[3], Vivian S W Li[2] & Barry J Thompson[1,5,*]

## Abstract

Wnt signalling induces a gradient of stem/progenitor cell proliferation along the crypt-villus axis of the intestine, which becomes expanded during intestinal regeneration or tumour formation. The YAP transcriptional co-activator is known to be required for intestinal regeneration, but its mode of regulation remains controversial. Here we show that the YAP-TEAD transcription factor is a key downstream effector of Wnt signalling in the intestine. Loss of YAP activity by *Yap/Taz* conditional knockout results in sensitivity of crypt stem cells to apoptosis and reduced cell proliferation during regeneration. Gain of YAP activity by *Lats1/2* conditional knockout is sufficient to drive a crypt hyperproliferation response. In particular, Wnt signalling acts transcriptionally to induce *YAP* and *TEAD1/2/4* expression. YAP normally localises to the nucleus only in crypt base stem cells, but becomes nuclear in most intestinal epithelial cells during intestinal regeneration after irradiation, or during organoid growth, in a Src family kinase-dependent manner. YAP-driven crypt expansion during regeneration involves an elongation and flattening of the Wnt signalling gradient. Thus, Wnt and Src-YAP signals cooperate to drive intestinal regeneration.

**Keywords** intestine; regeneration; Src; Wnt; YAP
**Subject Categories** Cancer; Development; Stem Cells & Regenerative Medicine
**The EMBO Journal (2021) 40: e105770**

## Introduction

The Wnt signalling pathway was discovered to signal via the beta-catenin transcriptional co-activator, which binds to the TCF/LEF1 family of DNA-binding transcription factors to control nuclear gene transcription (Bienz & Clevers, 2000; MacDonald *et al,* 2009; Clevers, 2013; Franz *et al,* 2017; Gammons & Bienz, 2018). In the mammalian intestine, Wnt ligands are secreted from mesenchymal niche cells (Valenta *et al,* 2016; Degirmenci *et al,* 2018) such that Wnt signalling forms a gradient along the crypt-villus axis, with beta-catenin/TCF-driven transcription strongest in the base of the crypt, where it induces expression of stem cell fate markers such as *Lgr5* and *Olfm4* (Bienz & Clevers, 2000; Clevers, 2013). In addition to controlling stem cell fate, one important function of the Wnt-induced beta-catenin/TCF activity gradient is to induce a corresponding gradient of cell proliferation to maintain normal intestinal homeostasis (Korinek *et al,* 1998; Ireland *et al,* 2004; Fevr *et al,* 2007; van Es *et al,* 2012; Valenta *et al,* 2016). Ectopic activation of beta-catenin, either directly or in *Apc* mutant intestinal cells, is sufficient to induce expanded hypertrophic proliferation along the crypt-villus axis or formation of adenomas (Korinek *et al,* 1997; Morin *et al,* 1997; Harada *et al,* 1999; Sansom *et al,* 2004; Andreu *et al,* 2005). One key target gene of beta-catenin/TCF is *Myc*, encoding a transcription factor expressed in a crypt-villus gradient that is required to promote cell proliferation during normal intestinal homeostasis and tumour formation (He *et al,* 1998; Muncan *et al,* 2006; Sansom *et al,* 2007; Finch *et al,* 2009). Another beta-catenin/TCF target gene, *Sox9*, is expressed similar gradient to *Myc* along the crypt-villus axis, but is also expressed in Paneth cells, where it is required for Paneth cell differentiation as marked by expression of *Lysozyme* (*Lyz*) (Bastide *et al,* 2007; Mori-Akiyama *et al,* 2007).

A remarkable feature of the intestine is its ability to regenerate after tissue damage, a process that involves a transiently expanded gradient of cell proliferation along the crypt-villus axis (Potten & Grant, 1998; Potten, 1998; Bach *et al,* 2000). Wnt/beta-catenin signalling, Myc and Sox9 are all essential for intestinal regeneration after damage, along with additional signalling proteins such as the focal adhesion kinase (FAK), Src kinase and the YAP transcriptional co-activator that are specifically required for regeneration (and organoid culture) but are dispensable for normal homeostasis (Ashton *et al,* 2010; Cai *et al,* 2010; Cordero *et al,* 2014; Gregorieff *et al,* 2015; Roche *et al,* 2015). Interestingly, FAK-Src-YAP form a

1 Epithelial Biology Laboratory, Francis Crick Institute, London, UK
2 Stem Cell and Cancer Biology Laboratory, Francis Crick Institute, London, UK
3 Colorectal Cancer and Wnt signalling Laboratory, Cancer Research UK Beatson Institute, Glasgow, UK
4 Institute of Cancer Sciences, Wolfson Wohl Cancer Research Centre, Bearsden, UK
5 EMBL Australia ACRF Department of Cancer Biology & Therapeutics, John Curtin School of Medical Research, The Australian National University, Acton, ACT, Australia
*Corresponding author. Tel: +61 2 61251068; E-mail: barry.thompson@anu.edu.au

signalling pathway that acts downstream of integrins to promote cell proliferation in skin (Kim & Gumbiner, 2015; Elbediwy *et al*, 2016; Li *et al*, 2016). During intestinal regeneration, Src phosphorylation is increased (Cordero *et al*, 2014) and the YAP protein becomes elevated and more strongly nuclear localised (Cai *et al*, 2010; Gregorieff *et al*, 2015). Recent work showed that FAK or Src inhibitors reduce nuclear YAP localisation during intestinal repair and confirmed the requirement for YAP in this process (Yui *et al*, 2018). Furthermore, activation of Src with overexpressed gp130 (IL6-ST) causes YAP to become nuclear localised throughout the crypt-villus axis (Taniguchi *et al*, 2015; Taniguchi *et al*, 2017).

The relationship between Wnt signalling and YAP in the intestine is a subject of some controversy. YAP protein is found at high levels in the crypts of the small intestine as well as in Apc mutant tumours or upon activation of beta-catenin (Cai *et al*, 2015; Gregorieff *et al*, 2015). Importantly, three groups demonstrated that conditional deletion of YAP abolished adenoma formation in *Apc*[Min] mice, indicating that YAP is essential for Wnt signalling to drive tumours (Azzolin *et al*, 2014; Cai *et al*, 2015; Gregorieff *et al*, 2015). However, YAP was dispensable for proliferative hypertrophy upon acute *Apc*[flox/flox] homozygous deletion throughout the intestine (Gregorieff *et al*, 2015; Taniguchi *et al*, 2015; Taniguchi *et al*, 2017), suggesting that hypertrophic growth is simply an expansion of the normal Wnt-dependent, YAP-independent, homeostatic proliferation programme, while tumour formation may require additional input from other signals that induce YAP nuclear translocation.

A prominent model has been proposed for the molecular mechanism connecting Wnt signalling with YAP in the gut: that Apc directly inactivates YAP via Axin-YAP binding to retain YAP in the cytoplasm and promote YAP degradation (Azzolin *et al*, 2014). This model proposes a post-translational mechanism for Wnt-dependent regulation of YAP subcellular localisation and stability, a notion that has been challenged by the observation that acute deletion of *Apc* in the intestine is not always sufficient to cause nuclear localisation of YAP, indicating that signals other than Wnt must drive YAP nuclear localisation during regeneration and tumour formation (Gregorieff *et al*, 2015; Gregorieff & Wrana, 2017). Alternative models suggest that YAP nuclear localisation in the intestine is inhibited by the canonical Hippo pathway (Cai *et al*, 2015; Gregorieff *et al*, 2015) and promoted by Src family

kinase signalling (Rosenbluh *et al*, 2012; Taniguchi *et al*, 2015; Taniguchi *et al*, 2017; Yui *et al*, 2018).

In addition to Wnt signalling promoting YAP function, there is evidence that YAP activity can induce "negative feedback" upon expression of certain Wnt target genes, such as *Lgr5, Olfm4* and *Lyz*—particularly during the intestinal regenerative response to irradiation (Gregorieff *et al*, 2015). Very recently, two groups reported that YAP activation can directly inhibit Wnt signalling in the intestine (Cheung *et al*, 2020; Li *et al*, 2020), with one report claiming that YAP therefore functions as a tumour suppressor, rather than an oncogene, in colorectal cancer (Cheung *et al*, 2020), building on their previous work (Barry *et al*, 2013), but in conflict with other reports of an oncogenic role for YAP in the intestine (Cai *et al*, 2015; Gregorieff *et al*, 2015). Thus, the role and regulation of YAP as a downstream effector of Wnt signalling is still not clearly resolved in either intestinal regeneration or cancer. We therefore sought to re-examine the relationship between Wnt signalling and YAP-TEAD activity in the intestine.

# Results

### Yap/Taz double knockouts exhibit crypt stem cell apoptosis and defects in the transverse folds of the ascending colon

We began by investigating the normal role of YAP/TAZ in the intestine by re-examining the consequences of loss of both proteins in *Villin-Cre*[ERt] *Yap*[flox/flox] *Taz*[flox/flox] double conditional knockouts (*Yap/Taz* dKO) treated with tamoxifen to induce deletion of both genes. Although YAP is far more strongly expressed than TAZ in the intestine (Fig EV1A–E), we induced deletion of both genes to be certain of a full loss of function. We confirm previous reports (Cai *et al*, 2010; Gregorieff *et al*, 2015) that loss of both YAP and TAZ does not strongly affect the overall morphology of the small intestine and detect only a mild increase in the rate of apoptosis in crypt base stem cells, while loss of both YAP and TAZ in the large intestine caused increased apoptosis, particularly in the transverse folds of the ascending colon (Fig 1A,B). Identical results were obtained using *Yap*[flox/flox] *Taz*[flox/flox] animals (i.e. without *Villin-Cre*[ERt] expression) or *Villin-Cre*[ERt] animals (i.e. without *Yap*[flox/flox] *Taz*[flox/flox]) as controls (Fig 1A–D). Notably, many *Yap/Taz* dKO animals showed profound defects in the transverse folds of the

---

**Figure 1. YAP/TAZ double knockouts exhibit defects in the transverse folds of the ascending colon.**

A Murine small intestines immunostained for the proliferation marker Ki67, apoptosis marker cleaved caspase 3 (Cas3) or YAP from four control (*Yap*[fl/fl] *Taz*[fl/fl]) and four YAP/TAZ double conditional knockout (*Villin-Cre*[ERt] *Yap*[fl/fl] *Taz*[fl/fl]) animals treated with tamoxifen. Tissues were harvested and analysed at day 7 after first tamoxifen injection. Loss of YAP/TAZ leads to a moderate increase in apoptosis (Cas3[+]) in crypt base stem cells, quantified on the right, but no overall effect on the morphology of the small intestine. Arrows point to apoptotic cells. *P < 0.05.

B Murine large intestines immunostained for proliferation marker Ki67, apoptosis marker cleaved caspase 3 (Cas3) or YAP from control (*Yap*[fl/fl] *Taz*[fl/fl]) and YAP/TAZ double conditional knockout (*Villin-CreERt Yap*[fl/fl] *Taz*[fl/fl]) animals treated with tamoxifen. Tissues were harvested and analysed at day 7 after first tamoxifen injection. Loss of YAP/TAZ leads to an increased number of apoptotic (Cas3[+]) cells within the transverse folds of the ascending colon. ****P < 0.0001.

C Murine small intestines immunostained for YAP or cleaved caspase 3 (Cas3) from control (*Villin-CreERt*) and YAP/TAZ double conditional knockout (*Villin-Cre*[ERt] *Yap*[fl/fl] *Taz*[fl/fl]) animals treated with tamoxifen. Tissues were harvested and analysed at day 7 after first tamoxifen injection. Loss of YAP/TAZ leads to a mild increase in apoptosis (Cas3[+]) in crypt base stem cells, but no overall effect on the morphology of the small intestine. Arrows point to stem cells (YAP nuclear staining), while arrows point to apoptotic cells (Cas3 staining). N > 10 animals per genotype.

D Murine large intestines immunostained for YAP or cleaved caspase 3 (Cas3) from control (*Villin-CreERt*) and YAP/TAZ double conditional knockout (*Villin-CreERt Yap*[fl/fl] *Taz*[fl/fl]) animals treated with tamoxifen. Tissues were harvested and analysed at day 7 after first tamoxifen injection. Loss of YAP/TAZ leads to increased apoptosis (Cas3[+]) and severe morphological defects in the transverse folds of the ascending colon, but does not affect morphology of the remaining colon regions. Quantitatively, 75% of n = 4 animals showed this severe phenotype.

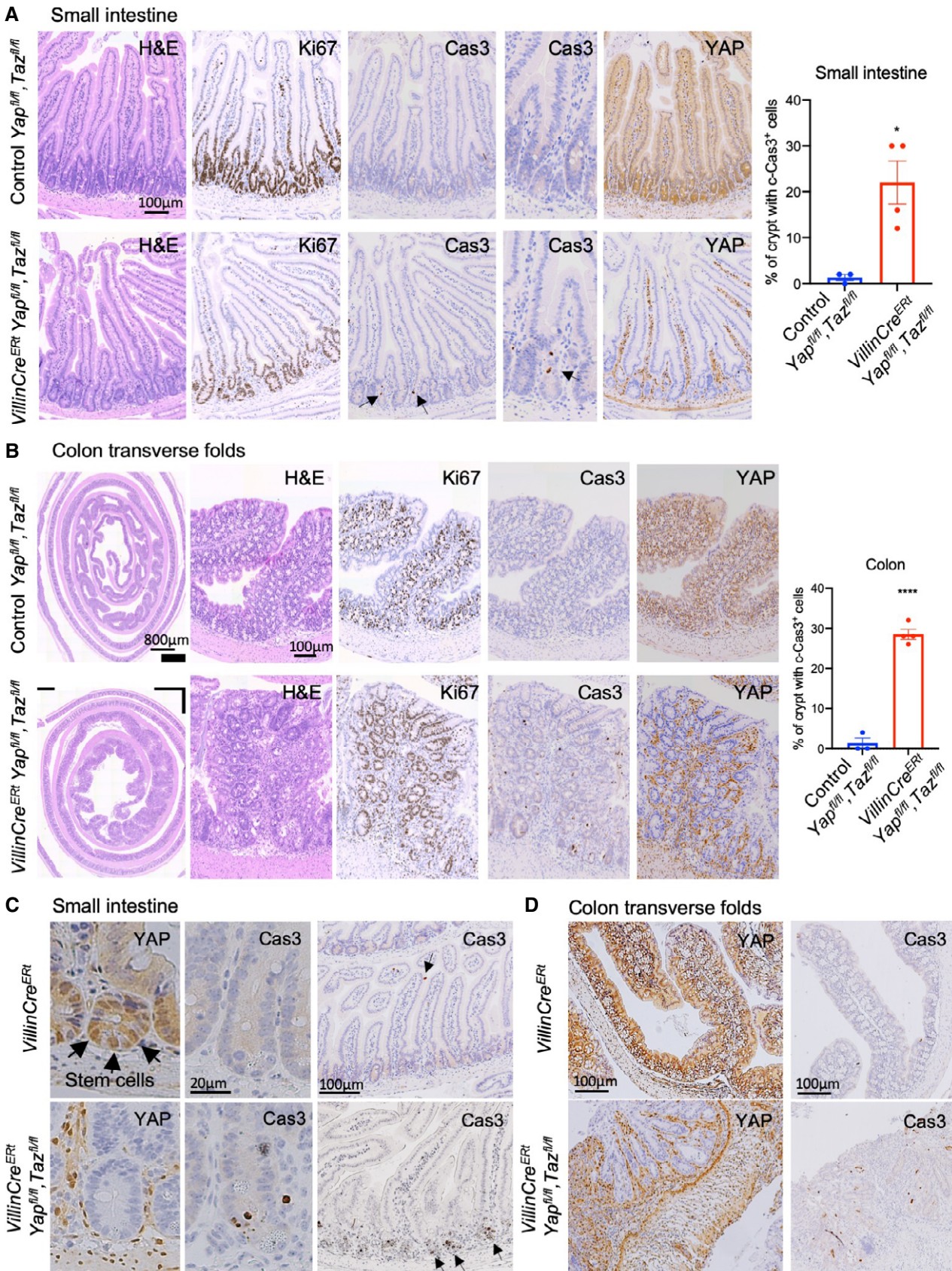

**Figure 1.**

ascending colon (75% of $n = 4$ animals; Fig 1D). These transverse folds project into the lumen of the colon and are therefore highly susceptible to mechanical damage, which may explain why YAP and TAZ are specifically required in these folds (Fig 1D). We confirm previous findings that after treatment with gamma irradiation (12 Gy), *Yap/Taz* dKO animals exhibit defective regeneration throughout the intestine (Fig EV2A and B, and EV3A and B). These findings identify a physiological requirement for YAP-TEAD signalling in promoting intestinal stem cell survival and in regeneration of regions susceptible to frequent tissue damage.

### *Lats1/2* dKO drives YAP nuclear localisation, crypt hyperplasia and a long/flat Wnt signalling gradient

We next examined the consequence of constitutively activating YAP-TEAD signalling in the intestine in *Villin-Cre$^{ERt}$ Lats1$^{flox/flox}$ Lats2$^{flox/flox}$* dKOs (*Lats1/2* dKO) treated with tamoxifen to induce deletion of both genes. We find that YAP becomes strongly nuclear localised throughout the intestinal epithelium, as expected, and that cell proliferation is strongly upregulated in crypts of both the proximal small and large intestines (Figs 2A and EV4A and B). Ectopic expression of *nlsYAP$^{5SA}$* causes a similar phenotype to *Lats1/2* dKO, mimicking an irradiation-induced regenerative state (Fig EV5A and B). Analysis of the Wnt target genes *Sox9* (detected by immunostaining) and *Axin2* (detected by RNA scope *in situ* hybridisation) reveals an expansion along the crypt-villus axis in *Lats1/2* dKO animals (Fig 2A–C), in contrast to recent reports that Wnt signalling is inactivated in these mutants (Cheung *et al*, 2020). The normally steep gradient of *Sox9* and *Axin2* expression is elongated and flattened in the *Lats1/2* dKO animals, indicating that the shape of the Wnt signalling gradient has been altered (Fig 2A–C). Expression of the high-threshold Wnt target gene and Paneth cell marker *Lyz* is also lost from the base of the crypts in *Lats1/2* dKO small intestines (Fig 2A). qPCR analysis confirms that high-threshold Wnt target genes such as *Lgr5* and *Olfm4* are reduced in *Lats1/2* dKO small intestines, while YAP targets such as *Ctgf* and *Cyr61* are strongly induced and the common Wnt and YAP target gene *CyclinD1* is also moderately induced (Fig 2D). These results indicate that nuclear localisation of YAP is sufficient to drive proliferation specifically in intestinal crypts, consistent with the notion that YAP must cooperate with Wnt signalling to drive the expression of crypt proliferation markers such as *Sox9* and *CyclinD1* (Fig 2A–C). These findings

agree with recent reports that *Myc* is also a common target gene of both YAP and Wnt signalling (Choi *et al*, 2018; Li *et al*, 2020). That the gradient of Wnt signalling undergoes elongation and flattening during YAP-driven crypt expansion (Fig 2C) explains the upregulation of transit-amplifying cell proliferation (and expansion of medium-threshold Wnt targets such as *Sox9, Myc* and *CyclinD1*) at the expense of crypt base stem cells (and expression of high-threshold Wnt targets such as *Lgr5* and *Olfm4*).

In the foregoing qPCR analysis, we noticed that expression of *TEAD1/4* mRNA (but not *TEAD2/3*) was dramatically induced in *Lats1/2* dKO small intestines, which we then confirmed by RNAscope *in situ* hybridisation to visualise single molecules in both the small intestine and colon (Fig 3A and B). Temporal analysis at 0, 3 and 7 days post-tamoxifen-induction (dpi) of *Lats1/2* dKO reveals the progressive increase in *TEAD1/4* mRNA expression, similar to that of the classical YAP target gene *Ctgf* (Fig 3C), while *TEAD2* exhibits a mild downregulation similar to the classical Wnt target gene *Axin2* (Fig 3C). These results show that *TEAD1* and *TEAD4* behave as classical YAP target genes, while *TEAD2* appears to behave as a classical Wnt target gene in the intestine.

### Wnt signalling induces *YAP* & *TEAD1/2/4* mRNA expression in the intestine

We next sought to directly test whether *TEAD1/4* and *TEAD2* are induced upon activation of Wnt signalling in *Apc$^{Min}$* mutant intestinal tumours. We find strong induction of all three genes in *Apc$^{Min}$* mutant tumours of both the small intestine and colon (Fig 4A). We confirmed induction of *TEAD1/2/4*, but not *TEAD3*, by qPCR analysis in *Apc* mutant intestinal organoids *in vitro*, *Apc$^{Min}$* mutant tumours *in vivo* and in *Apc$^{fl/fl}$* knockout intestinal crypts *in vivo* (Fig 4B). We further note that immunostaining for the TEAD4 protein in the Human Protein Atlas database reveals a gradient of expression along the crypt-villus axis in the human small intestine and colon, and is uniformly induced in human colorectal adenomas, similar to Sox9 (Fig 4C). Together, our results show that *TEAD1/4* are common Wnt and YAP target genes, while *TEAD2* is a Wnt-only target gene and *TEAD3* is regulated by neither pathway.

The above results raise the question of whether the *YAP* gene is also a target of Wnt signals in the intestine. We compared immunostaining with anti-YAP antibodies with *in situ* hybridisation for YAP mRNA in mouse small intestine (Appendix Fig S1A). We again took

**Figure 2.   LATS1/2 double knockout drives YAP nuclear localisation, crypt hyperplasia and a long/flat Wnt gradient.**

A    Murine small (top) and large (bottom) intestines isolated from control (Cre negative) *Lats1$^{flox/flox}$ Lats2$^{flox/flox}$* animals and *Villin-CreERt Lats1$^{flox/flox}$ Lats2$^{flox/flox}$* animals treated with tamoxifen to induce homozygous deletion of *Lats1/2* (dKO). Immunostaining for YAP and Ki67 shows a gradient of YAP expression along the crypt-villus axis in controls, with nuclear YAP- and Ki67-positive cells restricted to the crypt base (representative images from $n = 5$ mice). Tamoxifen-treated (3 days i.p.) *Villin-CreERt Lats1$^{flox/flox}$ Lats2$^{flox/flox}$* double homozygous mouse intestines show an enlarged crypt compartment after 7 days with strongly nuclear YAP immunostaining in all epithelial cells and an expanded proliferative zone marked by Ki67-positive cells. Note the gradient of YAP expression levels is maintained along the crypt-villus axis (representative images from $n = 5$ mice for each genotype). (Right) Immunostaining for the Paneth cell marker Lyz reveals loss of this marker from the crypt base. (Bottom right) qPCR analysis of Wnt pathway target genes reveal that *Lats1/2* dKO causes a mild reduction in *Lgr5* expression, with complete loss of *Olfm4*. $*P < 0.05$.

B    *Axin2* mRNA expression was measured by RNAscope (red) and found to be increased in both intensity and uniformity of staining along the crypt-villus axis in *Lats1/2* dKO after 3 days post-i.p. injection (dpi) with tamoxifen to induce the homozygous deletion of *Lats1* and *Lats2*. At 7 dpi, *Axin2* mRNA levels remain uniform along the crypt-villus axis in *Lats1/2* dKO animals, although their total level has declined compared to 3 dpi.

C    Schematic diagram showing alteration of the Wnt signalling gradient upon *Lats1/2* dKO in the intestine, with crypt expansion being accompanied by an elongation/ flattening of the Wnt signalling gradient, whose elevated level gradually declines over time.

D    qPCR analysis of YAP-TEAD and their target genes *Ctgf* and *Cyr61* reveal that *Lats1/2* dKO causes a strong induction of *Ctgf* and *Cyr61* expression, as expected, as well as a strong induction of the common Wnt and YAP target gene *CyclinD1*. $*P < 0.05$; $**P < 0.01$.

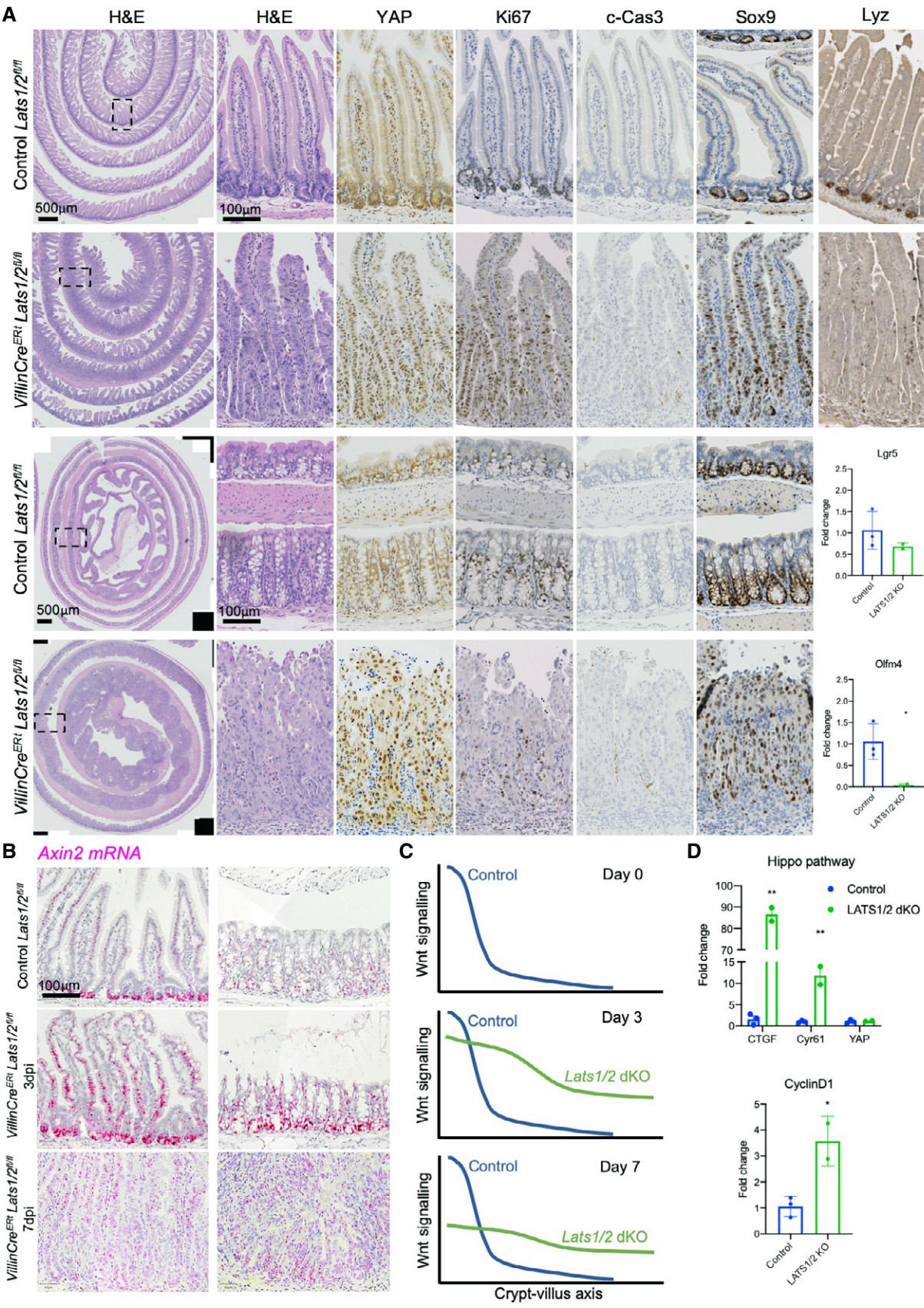

**Figure 2.**

advantage of the high sensitivity and specificity of the RNAscope technology to visualise single molecules of *YAP* mRNA. We find a correlation between the gradients of *YAP* mRNA and YAP protein level along the crypt-villus axis, with YAP most strongly expressed in the crypt (Appendix Fig S1A). These findings suggest that regulation of *YAP* gene expression is sufficient to explain the regulation of YAP protein levels by Wnt signalling. In support of this notion, activation of Wnt signalling in *Apc$^{fl/fl}$* mutant intestinal adenomas is sufficient to induce both *YAP* mRNA and YAP protein throughout the tumour (Appendix Fig S1B). Notably, most nuclei in the *Apc* mutant adenomas are negative for YAP protein, which remains largely cytoplasmic (Fig EV1C and Appendix Fig S1B). The subtle Wnt-induced increase in *YAP* mRNA expression between crypt and villus can be explained by beta-catenin/TCF4 activation of transcription from a single previously identified site in the YAP promoter (Konsavage *et al*, 2012) (Appendix Fig S1C–F). These results show that Wnt signalling can drive a subtle increase in the transcription of the *YAP* gene in the intestine, in addition to strong induction of *TEAD1/2/4* gene expression, indicating that the primary mechanism by which Wnt induces YAP-TEAD activity is transcriptional in nature.

To further test this hypothesis, we sought to examine YAP in mouse intestinal organoids upon Wnt pathway modulations. Inhibition of Wnt signalling by addition of Porcupine (Porc) inhibitor LGK974 or activation of Wnt signalling in *Apc* homozygous mutant (*Apc5*) organoids (Novellasdemunt *et al*, 2017) results in corresponding changes in *YAP* mRNA and protein levels (Fig 5A–C; Appendix Fig S1G and H). Notably, the magnitude of *YAP* mRNA regulation (Appendix Fig S1G and H) was very mild compared with the magnitude of *Ctgf or TEAD1/2/4* mRNA regulation in these experiments (Fig 4A–D, Appendix Figs S1G and H, and S2A), suggesting that strong upregulation of *TEAD1/2/4* mRNA levels by Wnt signalling may be of primary importance in determining YAP-TEAD activity. Notably, we confirm that expression of YAP is essential in either *Apc$^{fl/fl}$* or *Apc$^{fl/fl}$*, *p53$^{fl/fl}$* mutant tumour-derived organoids, which cannot grow when mutant for YAP/TAZ (Appendix Fig S1I). These results indicate that Wnt signalling acts transcriptionally to regulate expression of the *YAP* and *TEAD1/2/4* genes in the intestine, rather than acting post-translationally to regulate YAP localisation.

### YAP integrates Wnt signalling and mechanical stimuli in intestinal organoids

Recent reports indicate that YAP can become nuclear localised in organoids cultured in mechanically stiff matrix (Gjorevski *et al*, 2016), collagen matrix (Yui *et al*, 2018) or after 3 days of culture in soft Matrigel (Gregorieff *et al*, 2015). Accordingly, we find that YAP is generally initially cytoplasmic and then translocates to the nucleus in organoids after 3–5 days in Matrigel, when crypt buds form (Fig 5A). The inhibition of Wnt signalling by addition of Porcupine (Porc) inhibitor reduces YAP protein levels in crypt buds without having a strong effect on YAP nuclear localisation, presumably owing to continued expression of *TEAD3* (Fig 5B, Appendix Fig S2B–E). Activation of Wnt signalling in *Apc5* organoids is associated with generally increased nuclear YAP levels and consequent induction of *Ctgf mRNA* (Appendix Fig S2A), but also with a drastic change in morphology, such that the organoids grow to become highly cystic (Fig 5C) (Novellasdemunt *et al*, 2017). It is likely that mechanical stretching of the epithelium (stress and/or strain) contributes to the difference in YAP subcellular localisation in organoids, as in other epithelia (Zhao *et al*, 2007; Dupont *et al*, 2011; Wada *et al*, 2011; Benham-Pyle *et al*, 2015; Fletcher *et al*, 2018; Meng *et al*, 2018). In agreement with this view, in *Apc5* organoids, YAP is strongly nuclear in stretched cells yet cytoplasmic in cuboidal cells and the same regulation is observable in early wild-type organoids, prior to crypt budding, where small spheres have either a cuboidal or a stretched epithelium (Fig 5D and E). Thus, Wnt signalling transcriptionally controls YAP and TEAD levels, while YAP subcellular localisation is primarily regulated in a Wnt-independent fashion in intestinal organoids.

### YAP nuclear localisation is regulated by Src family kinase activity

We next sought to clarify which alternative signalling mechanisms might drive translocation of YAP from the cytoplasm to the nucleus upon organoid culture. One of the main candidates promoting YAP nuclear localisation is the Src family kinases (Cordero *et al*, 2014; Kim & Gumbiner, 2015; Elbediwy *et al*, 2016; Li *et al*, 2016; Si *et al*, 2017), which can also become activated in wild-type or *Apc* mutant intestine (Cordero *et al*, 2014) via inflammatory cytokine signalling or experimentally induced colitis (Taniguchi *et al*, 2015; Taniguchi *et al*, 2017; Yui *et al*, 2018). We therefore treated intestinal organoids featuring nuclear YAP, with the Src family kinase inhibitor Dasatinib. We find that Dasatinib efficiently blocks the normal nuclear localisation of YAP in organoid buds, without affecting the gradient of YAP expression (Fig 6A and B). Comparably, YAP becomes cytoplasmic (and less active) throughout the whole epithelium in all *Apc5* organoids treated with Dasatinib, or with the more specific Src inhibitor eCF506, indicating that YAP nuclear localisation is Src-dependent even in Wnt active conditions (Fig 6A and C).

---

**Figure 3.** *TEAD1/4 mRNA expression is progressively expanded in Lats1/2 dKO intestines.*

A   qPCR analysis of *TEAD1-4* mRNA expression in control versus *Lats1/2* dKO intestines at 7 days post-i.p. injection with tamoxifen. Note the strong increase in *TEAD1* and *TEAD4* mRNA levels in particular, which indicates that these are both target genes of YAP signalling. **$P < 0.01$.

B   RNAscope *in situ* hybridisation analysis of *TEAD1/2/4* mRNA expression from control (Cre negative) *Lats1$^{flox/flox}$ Lats2$^{flox/flox}$* animals and *Villin-CreERt Lats1$^{flox/flox}$ Lats2$^{flox/flox}$* animals treated with tamoxifen to induce homozygous deletion of *Lats1/2* (dKO). Tamoxifen-treated (3 days i.p.) *Villin-CreERt Lats1$^{flox/flox}$ Lats2$^{flox/flox}$* double homozygous mouse intestines show elevated expression of *TEAD1* and *TEAD4* in both the small intestine (top) and colon (bottom). Tissues were harvested and analysed at day 7 after first tamoxifen injection. ($n$ = 5 animals for each genotype).

C   qPCR analysis reveals a progressive increase in the YAP-TEAD target gene *Ctgf* in *Lats1/2* dKO intestines at 3 and 7 days post-i.p. injection with tamoxifen, as well as similarly increased *TEAD1* and *TEAD4* expression, confirming that *TEAD1* and *TEAD4* are YAP-responsive genes. Notably, *Axin2* and *TEAD2* exhibit a mild but progressive decline over 3–7 days after *Lats1/2* deletion, consistent with *Axin2* being a known Wnt-specific target gene and with Wnt signalling remaining active in *Lats1/2* dKO intestines. *$P < 0.05$; **$P < 0.01$; ***$P < 0.001$.

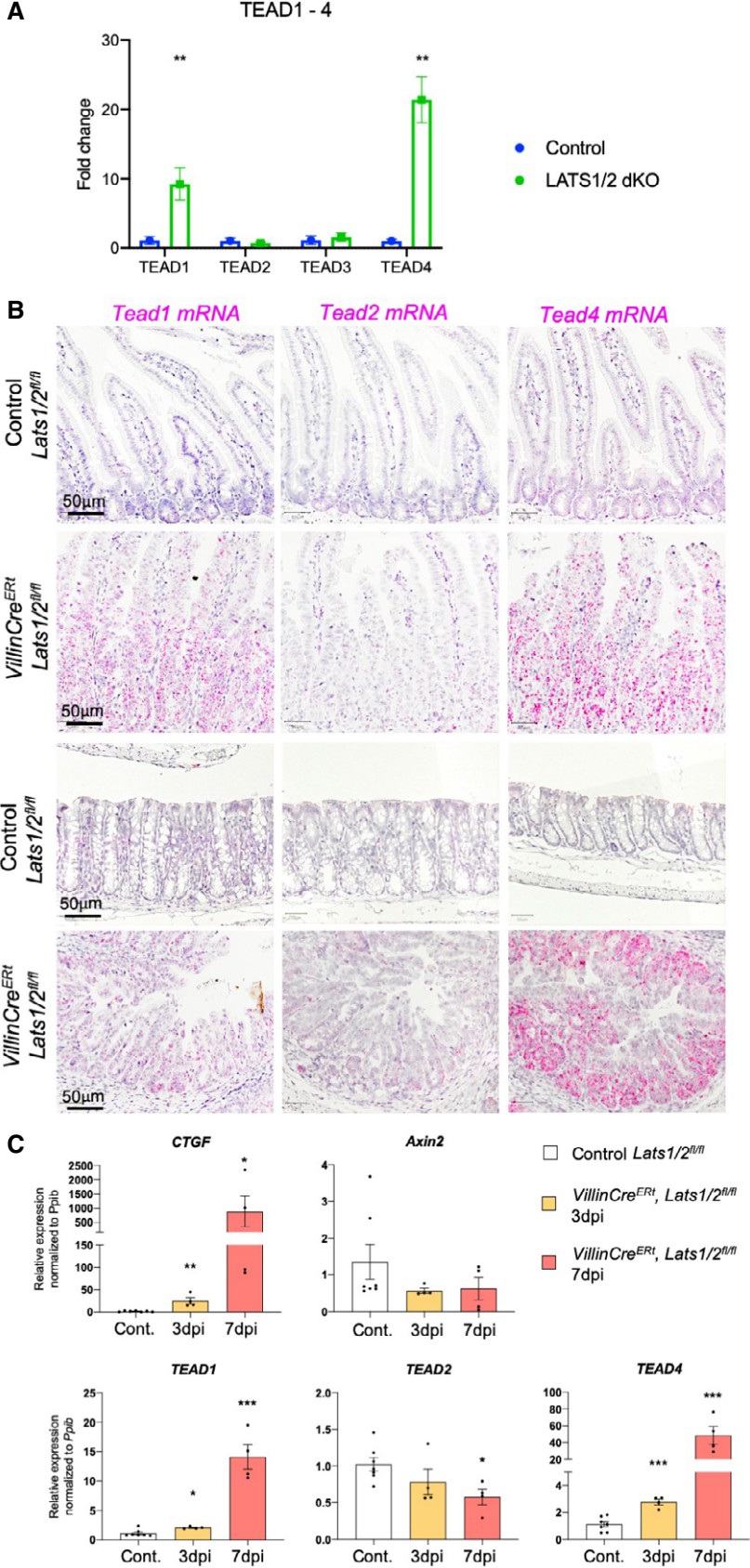

Figure 3.

These results indicate that Src family kinase activity is required for nuclear localisation of YAP in organoids in response to mechanical cues arising during growth of organoids into spherical cysts or during crypt budding.

Previous work in cell culture has produced two different models for how Src family kinases might regulate YAP. Src activation has been reported to drive YAP to the nucleus either via direct tyrosine phosphorylation of YAP or via phosphorylation of LATS1 (Li *et al*, 2016; Si *et al*, 2017). In support of the latter hypothesis, knockouts for MST1/2, Sav1 or LATS1/2 in the intestine are sufficient to induce YAP nuclear localisation (Cai *et al*, 2010; Zhou *et al*, 2011; Cai *et al*, 2015; Gregorieff *et al*, 2015) and we find that *Lats1/2* dKO intestines exhibit a crypt expansion phenotype with elevated cell proliferation (Fig 2A). We therefore tested whether Src inhibition with Dasatinib would have any effect in *Lats1/2* dKO organoids. We find that YAP becomes strongly nuclear in all cells of the *Lats1/2* dKO intestines and organoids, as expected, and that treatment with Dasatinib can only partially reduce YAP nuclear localisation in this context, with eCF506 having no effect (Fig 6D). These results support the notion that Src family kinases act primarily via inhibition of LATS1/2 (Si *et al*, 2017) rather than via a LATS-independent mechanism, such as direct tyrosine phosphorylation of YAP (Taniguchi *et al*, 2015; Li *et al*, 2016; Elbediwy *et al*, 2018). Thus, our findings support the view that YAP subcellular localisation is primarily controlled by the opposing action of LATS1/2 and Src family kinase signalling, independently of the Wnt pathway.

### Irradiation-induced nuclear localisation of YAP requires Src family kinase signalling

Interestingly, Src was recently shown to become strongly active after irradiation in the intestinal epithelium where it is required for a proper regeneration (Cordero *et al*, 2014). Since the intestinal-specific *Yap* single knockout and *Yap, Taz* double knockout each impair intestinal regeneration (Cai *et al*, 2010; Gregorieff *et al*, 2015; Gregorieff & Wrana, 2017), we wondered whether activated Src could be required to drive nuclear YAP during irradiation-induced regeneration. We find that Src family kinase activity is also required for the strong nuclear localisation of YAP that occurs following irradiation both in organoids and *in vivo* (Fig 7A and B) (Elbediwy *et al*, 2016). Notably, the nuclear translocation of YAP occurs within 4 h of irradiation in freshly cultured organoids, which are cultured in the absence of inflammatory immune cells or stromal tissue, suggesting that Src family kinases and YAP can act directly within the epithelium itself as sensors of DNA damage, in addition to their established roles as sensors of mechanical forces (Kim & Gumbiner, 2015; Elbediwy *et al*, 2016; Si *et al*, 2017; Elbediwy *et al*, 2018) or

cytokine/prostaglandin signalling (Taniguchi *et al*, 2015; Taniguchi *et al*, 2017; Roulis *et al*, 2020) (Fig 7A).

### Irradiation-induced YAP-TEAD activity and crypt regeneration is Wnt-dependent

Given that Src-YAP signalling becomes strongly active upon irradiation of the intestinal epithelium, we sought to examine the consequences for YAP and Wnt target gene expression and the role of Wnt signals. We find that, as expected, expression of both *Sox9* (analysed by immunohistochemistry) and *Myc* (analysed by RNAscope *in situ* hybridisation) becomes expanded after irradiation, consistent with the notion that Wnt signalling is still active (analysed by nuclear beta-catenin staining) and that *Sox9* & *Myc* are common YAP and Wnt target genes (Fig 8A). qPCR analysis confirms that YAP-only target genes (*Ctgf* & *Cyr61*) are strongly induced upon irradiation, while a Wnt-only target gene (*Axin2*) is mildly reduced. Other common YAP and Wnt targets show intermediate effects, with *TEAD4* being strongly induced while *TEAD1* and *Sox9* levels being only moderately altered (Fig 8B). Importantly, the expansion of the Sox9 expression domain upon irradiation does require active Wnt signalling, as it is strongly reduced in size, and *Sox9* mRNA levels decrease, upon treatment with Porc inhibitor (Fig 8C and D). Interestingly, p-Src levels are also reduced upon treatment with Porc inhibitor (Fig 8C), possibly reflecting a role of the Wnt target gene *CD44* in promoting Src activation and LATS1/2 inhibition in crypts, similar to its function in other cell and tumour types (Bourguignon *et al*, 2001; Li *et al*, 2001; Xu *et al*, 2010; Nam *et al*, 2015; Pastushenko *et al*, 2021). Indeed, intestinal knockouts of *CD44* cause a phenotype similar to the *Src* KO or *Yap/Taz* dKO, featuring increased crypt apoptosis and prevention of *Apc^Min* tumour formation (Zeilstra *et al*, 2008). These findings are consistent with an elongation and flattening of the Wnt gradient as the crypt expands upon irradiation, with an essential role of the re-shaped Wnt signalling gradient in driving regeneration in combination with YAP-TEAD activation. The essential role of Wnt signalling in supporting YAP-TEAD expression and target gene activation is also evident by qPCR analysis of crypts *in vivo* and in organoids after irradiation, where expression of both Wnt and YAP target genes is strongly affected by treatment with Porc inhibitor (Appendix Fig S2E and F). Together, these results show that intestinal regeneration is orchestrated by the combined action of YAP-TEAD signalling with a re-shaped Wnt signalling gradient.

Our results argue against the notion that YAP-TEAD is a direct inhibitor of Wnt signalling in the intestine (Cheung *et al*, 2020; Li *et al*, 2020), which led one group to propose that YAP function as a tumour suppressor (Cheung *et al*, 2020), building on their previous

---

**Figure 4. Wnt signalling induces *TEAD1/2/4* mRNA expression in the intestine.**

A    RNAscope in situ hybridisation analysis of *TEAD1/2/4* mRNA expression from control (wild type) and *Apc^Min* mutant tumours from the small intestine (top) and colon (bottom). All three genes were strongly induced within the *Apc^Min* mutant tumours, with *TEAD2* being most strongly induced in colon *Apc^Min* tumours (*n* = 3 animals with multiple tumours within the small and large intestines).

B    qPCR analysis of control and *Apc5* mutant murine intestinal organoids *in vitro*, control and *Apc^Min* mutant tumour samples *in vivo*, and control and *Apc^fl/fl* knockout hyperplastic crypts *in vivo*. In all cases, *TEAD1/2/4* mRNA expression was strongly elevated upon loss of Apc function, while *TEAD3* mRNA expression was unaffected. (RNA was isolated from six individual tumours). **P* < 0.05; *****P* < 0.0001.

C    Antibody staining for TEAD4 reveals a gradient of expression along the crypt-villus axis in both the small and large intestine, with uniformly high expression in colorectal adenomas, similar to SOX9.

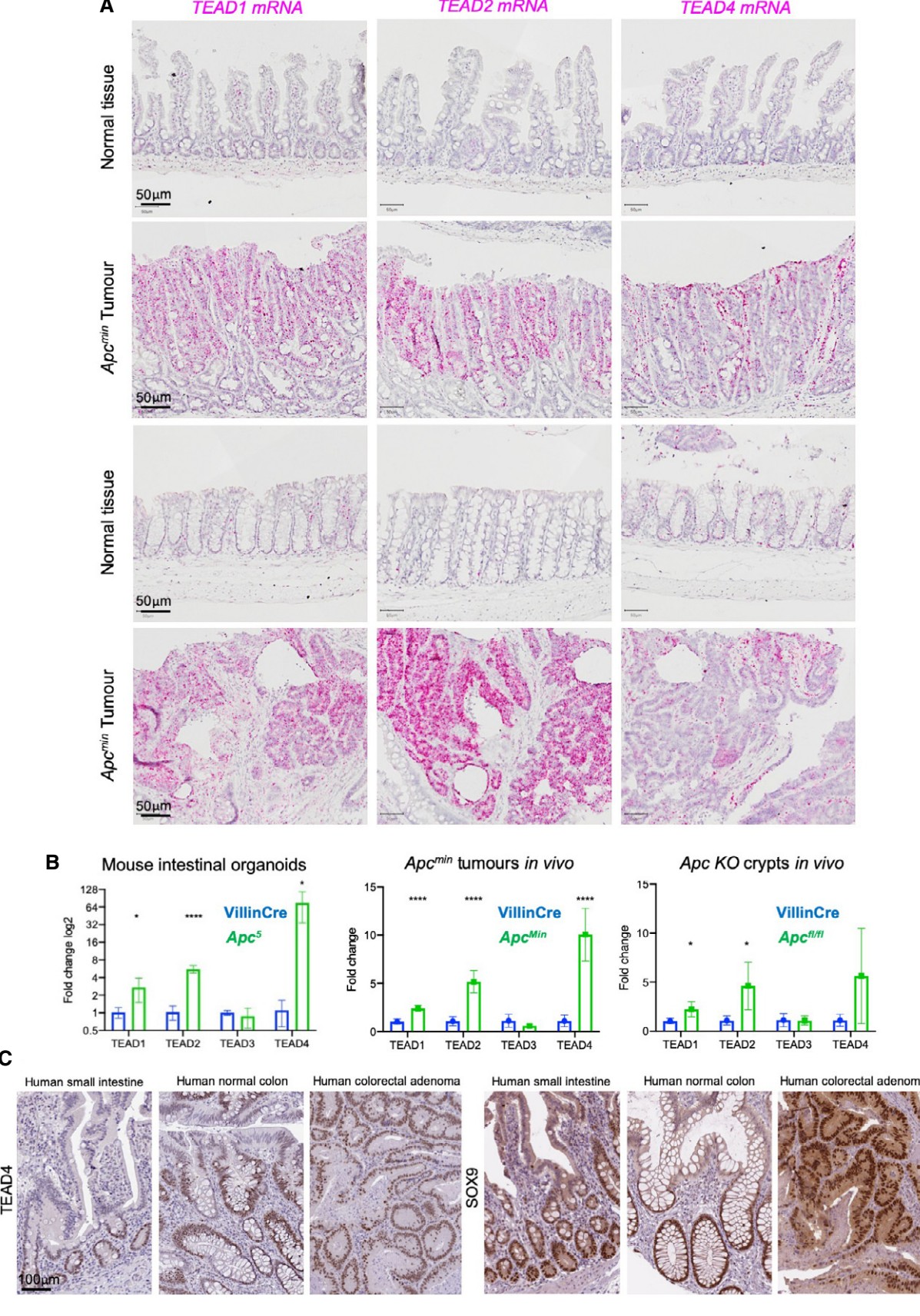

**Figure 4.**

findings that activation of a *tetO-YAP^{S127A} rtTA* transgene inhibits intestinal proliferation (Barry *et al*, 2013). We find that YAP activation (with either *Lats1/2* dKO or transgenic *nlsYAP^{SSA}*) does not inhibit cell proliferation or abolish Wnt signalling *in vivo* (Figs 2, 3 and EV3–EV5). Furthermore, *Lats1/2* dKO organoids grow normally and express Sox9 (Appendix Fig S3), while expression of transgenic *nlsYAP^{SSA}* does not impair growth of *Apc^{fl/fl} p53^{fl/fl}* knockout organoids *in vitro* (Appendix Fig S4) or formation of tumours after subcutaneous implantation into nude mice (Appendix Fig S5). These results support the notion that YAP acts as a pro-proliferative and oncogenic factor during intestinal regeneration and tumour progression.

## Discussion

Our findings confirm that YAP is normally nuclear localised in crypt base stem cells and show that *Yap/Taz* dKOs (*Yap/Taz* dKO) render stem cells susceptible to apoptosis. In addition, *Yap/Taz* dKO intestines revealed defects in the transverse folds of the ascending colon, a region exposed to a higher likelihood of tissue damage. These findings confirm a physiological requirement for the role of YAP in maintaining cell survival and promoting proliferative regeneration—a function previously observed after experimentally induced intestinal damage by irradiation or dextran sodium sulphate (DSS) treatment, which drives strong YAP nuclear localisation throughout the intestine (Cai *et al*, 2010; Gregorieff *et al*, 2015), which we also confirm. Ectopic induction of YAP nuclear localisation in *Lats1/2* dKO intestines was sufficient to drive crypt expansion via hyperproliferation of transit-amplifying cells, in agreement with recent findings (Li *et al*, 2020). The *Lats1/2* dKO phenotype is highly similar to that caused by conditional expression of *nlsYAP^{SSA}*, or that occurring during regeneration after irradiation, indicating that YAP activation is both necessary and sufficient to drive the regenerative response.

Our results also show that the *YAP* and *TEAD1/2/4* genes are regulated transcriptionally by Wnt/beta-catenin signalling in the intestine, while nuclear translocation of YAP upon tissue damage depends on Src family kinase signalling. The Wnt-induced transcriptional gradient of *YAP mRNA* expression is sufficient to explain the resulting gradient of YAP protein along the crypt-villus axis, as well as the elevation of YAP protein observed in *Apc* mutant tumours, while the Wnt-induced *TEAD1/2/4 mRNA* expression is even more potent. Src-dependent YAP nuclear localisation is then necessary for activation of YAP-TEAD transcriptional targets and transit-amplifying cell proliferation (see graphical abstract for a schematic model). Consequently, YAP remains mostly cytoplasmic until the tissue undergoes a regenerative response, when Src family kinases inhibit LATS1/2 and thereby drive YAP to the nucleus to enable it to activate YAP-TEAD-mediated transcription.

These results explain why Src-YAP signalling is largely dispensable for normal intestinal development but essential to prime the intestinal regeneration response (Ashton *et al*, 2010; Cai *et al*, 2010; Cordero *et al*, 2014; Gregorieff *et al*, 2015; Taniguchi *et al*, 2015). Notably, the phenotype of *Lats1/2* dKO, or *nlsYAP5SA* expression, appears to be a chronic regenerative proliferation response, which we observe induces diarrhoea, strongly reminiscent of inflammatory colitis—a key factor predisposing patients to colorectal cancer. Importantly, uniform nuclear YAP does not drive proliferation in differentiated cells of the villus, presumably because these villar cells lack sufficient Wnt signalling to induce *TEAD1/2/4* expression, or to induce common Wnt and YAP target genes such as *Sox9* and *Myc*, which explains why regenerative proliferation is normally limited to transit-amplifying cells of the intestinal crypt. In contrast, in *Apc* mutant tumour cells, uniformly elevated Wnt signalling induces *YAP*, *TEAD1/2/4*, *Sox9* and *Myc* expression throughout the tissue, enabling all cells to proliferate in an unlimited fashion in response to conditions that activate YAP, which include various stimuli such as DNA damage, mechanical stress or inflammatory signals (Thompson, 2020). Thus, all of our results are consistent with a pro-proliferative, pro-inflammatory and oncogenic role for YAP in both regeneration and in *Apc* mutant tumours, in agreement with several other reports (Cai *et al*, 2010; Cai *et al*, 2015; Gregorieff *et al*, 2015; Taniguchi *et al*, 2015; Taniguchi *et al*, 2017; Roulis *et al*, 2020).

Very recently, Li *et al* (2020) and Cheung *et al* (2020) reported that activation of YAP *directly inhibits* Wnt signalling in the intestinal epithelium, with one of these reports claiming that YAP therefore functions as a "tumour suppressor" in colorectal cancer (Cheung *et al*, 2020), in line with previous work from the same group (Barry *et al*, 2013). Our findings are only partly consistent with these reports. Since YAP is normally localised to the nucleus in crypt base stem cells, all groups agree that physiological YAP activity does not normally interfere with intestinal homeostasis, Wnt signalling or expression of high-threshold Wnt target genes: crypt stem cell markers (*Lgr5*, *Olfm4*) or Paneth cell markers (*Lyz*) (Cai *et al*, 2010; Gregorieff *et al*, 2015; Cheung *et al*, 2020; Li *et al*, 2020). Furthermore, in *Yap/Taz* dKO, there is no observable increase in

**Figure 5. YAP integrates Wnt signalling and mechanical stimuli in intestinal organoids.**

A   YAP and Ezrin immunostaining in control murine intestinal organoids cultured for 1–5 days as indicated. Note the initially cytoplasmic localisation of YAP, which becomes nuclear localised after 3 days.

B   YAP immunostaining of 3-day-old wild-type organoids treated with DMSO (Control) or Porcupine inhibitor LGK974 at 5 μM. Wnt signalling inhibition upon Porcupine inhibitor treatment decreases YAP levels without affecting YAP subcellular localisation. Quantification of YAP protein levels is shown below, with statistical significance in an unpaired two-tailed *t*-test. ****$P < 0.0001$.

C   Wnt activation in *Apc5* homozygous mutant organoids drives a drastic shift in organoid morphology from budding structures to cystic spheres and increases the YAP protein level. Scale bar is 250 μm. $n > 10$ organoids in each experiment. Quantification of YAP protein levels from crypt and villus regions of control organoids as well as from the regions bounded by rectangles in *Apc5* organoids are shown on the right, with statistical significance in an unpaired two-tailed *t*-test. ****$P < 0.0001$.

D   Wild-type organoids, at early stages of development (1–2 days) prior to crypt budding, form small spheroids that also exhibit either a columnar or a stretched epithelium where YAP is cytoplasmic or nuclear, respectively.

E   *Apc5* organoids (5 days) exhibit either a columnar or a stretched epithelium where YAP is cytoplasmic or nuclear, depending on the shape of the cells.

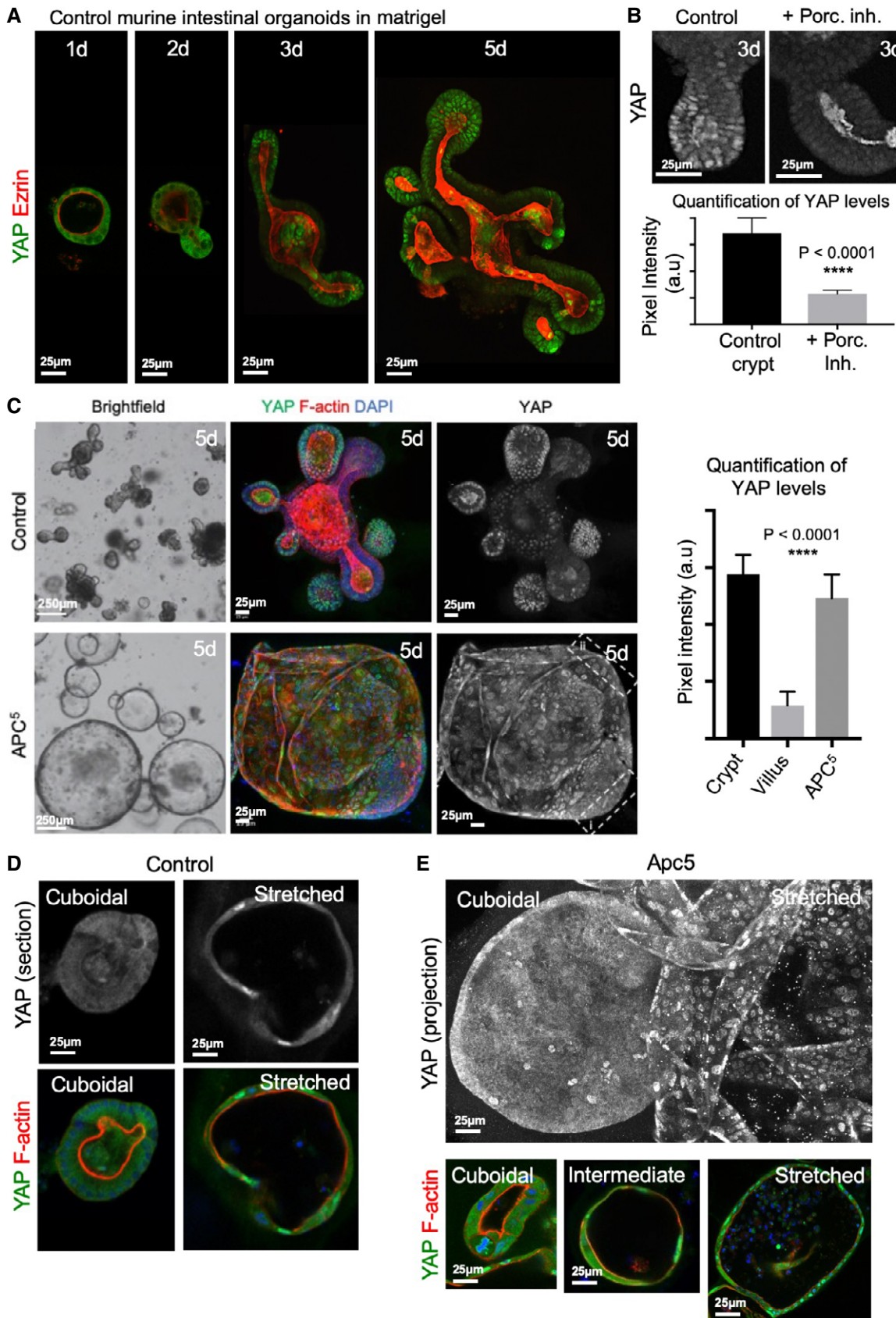

**Figure 5.**

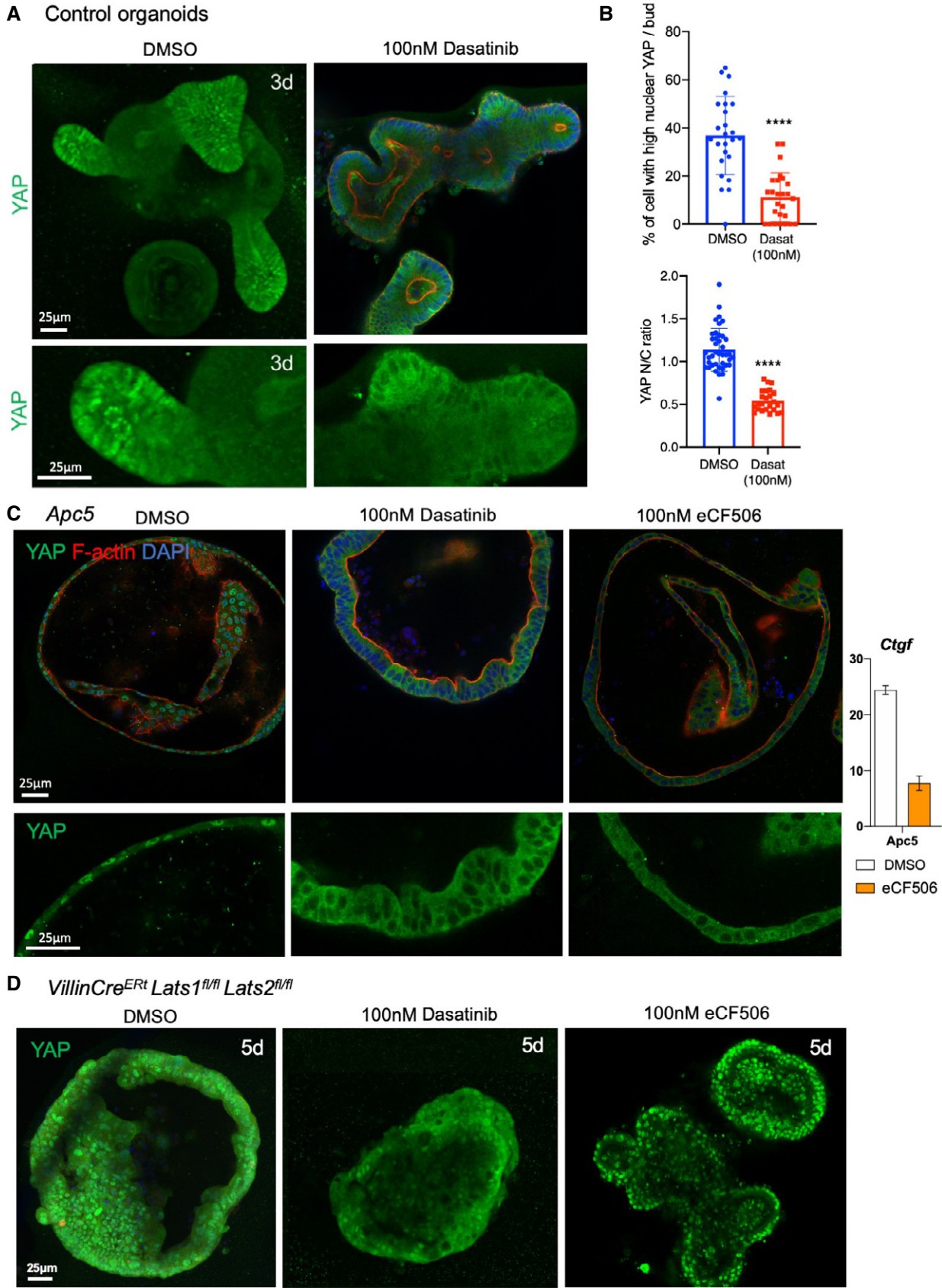

Figure 6.

**Figure 6.   In both wild-type and *Apc5* organoids, YAP nuclear localisation is prevented by a Src family kinase inhibitor.**

A   Src family kinase inhibition via Dasatinib treatment in control organoids enforces YAP cytoplasmic localisation after 4 h of treatment in organoids cultured for 3 days. *n* > 10 organoids in each experiment.

B   Quantification of the percentage of nuclear YAP cells per control crypt bud as shown in panel (A), error bars show 1 SD (****P < 0.0001). *n* > 10 organoids per condition (top). Quantification of the nuclear/cytoplasmic ratio of YAP per cell in *Apc5* mutant organoids as shown in panel (C). *n* > 10 organoids per condition (bottom). Statistical analysis was performed with an unpaired Student's *t*-test.

C   Src family kinase inhibition via Dasatinib or eCF506 treatment in *Apc5* organoids causes relocalisation of YAP to the cytoplasm. *n* > 10 organoids in each experiment. qPCR analysis of the YAP-TEAD target gene *Ctgf* reveals a strong inhibition of mRNA expression upon treatment with eCF506 (relative expression, error bars = 1 SD). Statistical analysis was performed with an unpaired Student's *t*-test.

D   Src family kinase inhibition via Dasatinib or eCF506 treatment in *Villin-Cre^ERt Lats1/2* dKO organoids (induced *in vitro* with 4-OHT) fails to cause relocalisation of YAP to the cytoplasm after 4 h treatment in culture. *n* > 10 organoids in each experiment.

Source data are available online for this figure.

Wnt target gene expression in the intestinal epithelium (Cai *et al*, 2010; Gregorieff *et al*, 2015; Li *et al*, 2020). Thus, any potential role of YAP as an inhibitor of Wnt signalling does not manifest during normal stem cell proliferation or intestinal homeostasis—arguing against any general function for YAP as a Wnt signalling inhibitor in the intestine, as variously proposed to occur via inhibition of Dishevelled (DVL) function (Varelas *et al*, 2010; Barry *et al*, 2013), direct binding and retention of beta-catenin in the cytoplasm (Imajo *et al*, 2012) or incorporation of YAP into the beta-catenin destruction complex (Azzolin *et al*, 2014).

In contrast, the Wnt signalling gradient could be indirectly affected by the YAP-driven crypt hyperproliferation phenotype during regeneration. Wnt gradients are known to "expand" to adjust to the size of the tissue over which they spread (Capek & Muller, 2019). Indeed, the rapid proliferation, migration (Krndija *et al*, 2019) and turnover of cells along the crypt-villus axis during regeneration would be expected to carry secreted Wnt molecules across a greater distance than in normal homeostasis, elongating and flattening the Wnt signalling gradient. According to this "gradient scaling" model, expression of high-threshold Wnt target genes (crypt stem cell markers) would be lost while expression of medium-threshold Wnt target genes would actually be expanded along the crypt-villus axis. Consistent with this model, Li *et al* and Cheung *et al* show that expression of high-threshold Wnt target genes *Lgr5*, *Olfm4* & *Lyz* is reduced in *Lats1/2* cKO intestines, owing to activation of YAP (Cheung *et al*, 2020; Li *et al*, 2020), results supported by our own analysis in these mutants (Fig 2A–D). Importantly, medium-threshold Wnt target genes, such as *Axin2*, *Sox9* and *Myc* (Li *et al*, 2020), are actually induced and expanded upon YAP activation in *Lats1/2* dKO intestines (Fig 2A), with *Sox9* and *Myc* (common Wnt and YAP target genes) being more highly expressed than *Axin2* (a Wnt-only target gene), whose expression gradually declines. Thus, our results

support an elongation/flattening of the Wnt signalling gradient upon YAP activation in the intestine, rather than a direct inhibition of Wnt signalling by YAP in this tissue.

Nevertheless, we do not rule out an eventual "negative feedback" role for YAP on Wnt signalling that emerges gradually during the intestinal regenerative response, which could contribute to the loss of high-threshold Wnt target genes such as *Lgr5*, *Olfm4* and *Lyz* (Gregorieff *et al*, 2015) and to the gradual decline in *Axin2* levels observed in *Lats1/2* dKO intestines. *Klf6* (Cheung *et al*, 2020) or *Clu* (Ayyaz *et al*, 2019) are examples of regeneration-specific YAP-TEAD target genes, although whether they can inhibit Wnt signalling remains unknown. *Nkd*, a gene induced during intestinal regeneration (Van Landeghem *et al*, 2012), is a well-known inhibitor of Wnt signalling that interferes with DVL function (Zeng *et al*, 2000; Gammons *et al*, 2020) and could provide a more plausible mechanism if found to be a YAP target gene in this tissue. *Dkk1*, also induced upon intestinal inflammation, is another well-known inhibitor of Wnt signalling and could conceivably be induced downstream of YAP (Pinto *et al*, 2003; Kuhnert *et al*, 2004; Nava *et al*, 2010; Koch *et al*, 2011).

In summary, our findings help resolve conflicting models of how regenerative proliferation of intestinal crypts occurs after tissue damage and of the role of YAP-TEAD and their regulation by Wnt signalling during this process. Our findings show that in addition to the well-known function of Wnt signalling in maintaining homeostatic stem/progenitor cell proliferation and controlling cell fate along crypt-villus axis, the Wnt pathway also primes the intestinal crypt for regeneration via inducing a gradient of *YAP*, *TEAD1/2/4* and *CD44* (which recruits Src). Upon tissue damage, strong Src activation and consequent YAP nuclear translocation drive YAP-TEAD-mediated transcription to increase cell proliferation to promote rapid regeneration. The crypt

**Figure 7.   Irradiation-induced nuclear localisation of YAP involves Src family kinase signalling.**

A   Mouse intestinal organoids immunostained for YAP (green). Note the mostly cytoplasmic localisation of YAP at 48 h in the organoid. Irradiation with 4 Gy of X-rays drives strong nuclear localisation of YAP in many cells, which is reversible by treatment with the Src family kinase inhibitor Dasatinib. The Dasatinib-treated organoids exhibit increased cell death after irradiation, as indicated by pyknotic nuclei (white arrow). *n* > 10 organoids in each experiment. This experiment employs the same control batch of 2–3 days organoids also shown in Fig 6A.

B   Control *Villin-Cre^ERt* mouse small intestine immunostained for YAP shows a gradient of expression with mostly cytoplasmic localisation. Irradiation with 14 Gy of X-rays drives strong nuclear localisation of YAP along the entire crypt-villus axis. Deletion of Src within *Villin-Cre^ERt Src^flox/flox* intestines prevents YAP nuclear localisation and normal regeneration after 14 Gy irradiation. Arrows point to villar cells with nuclear YAP localisation. Lower panels show high-mag views of a different example than those shown in upper panels.

Source data are available online for this figure.

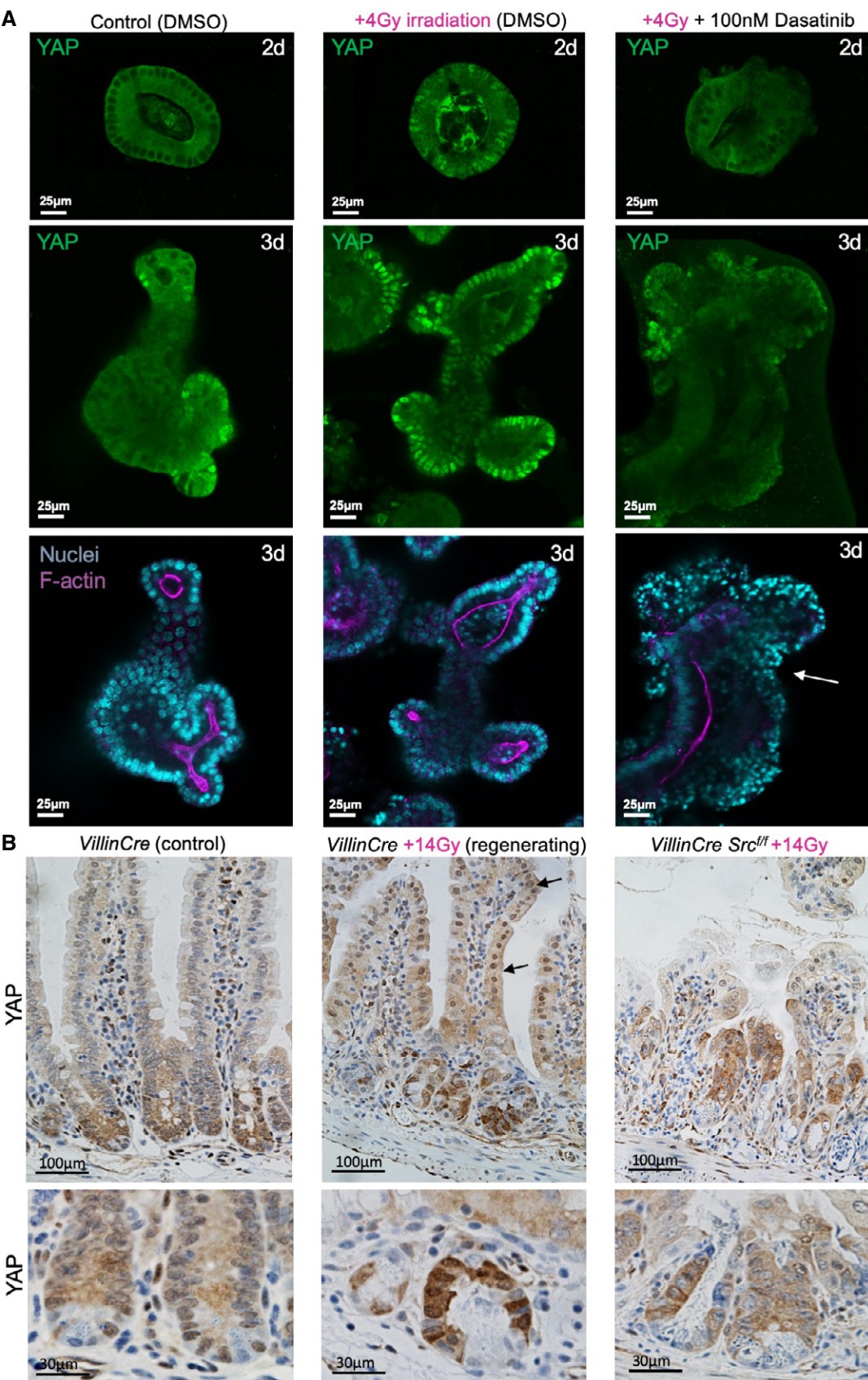

**Figure 7.**

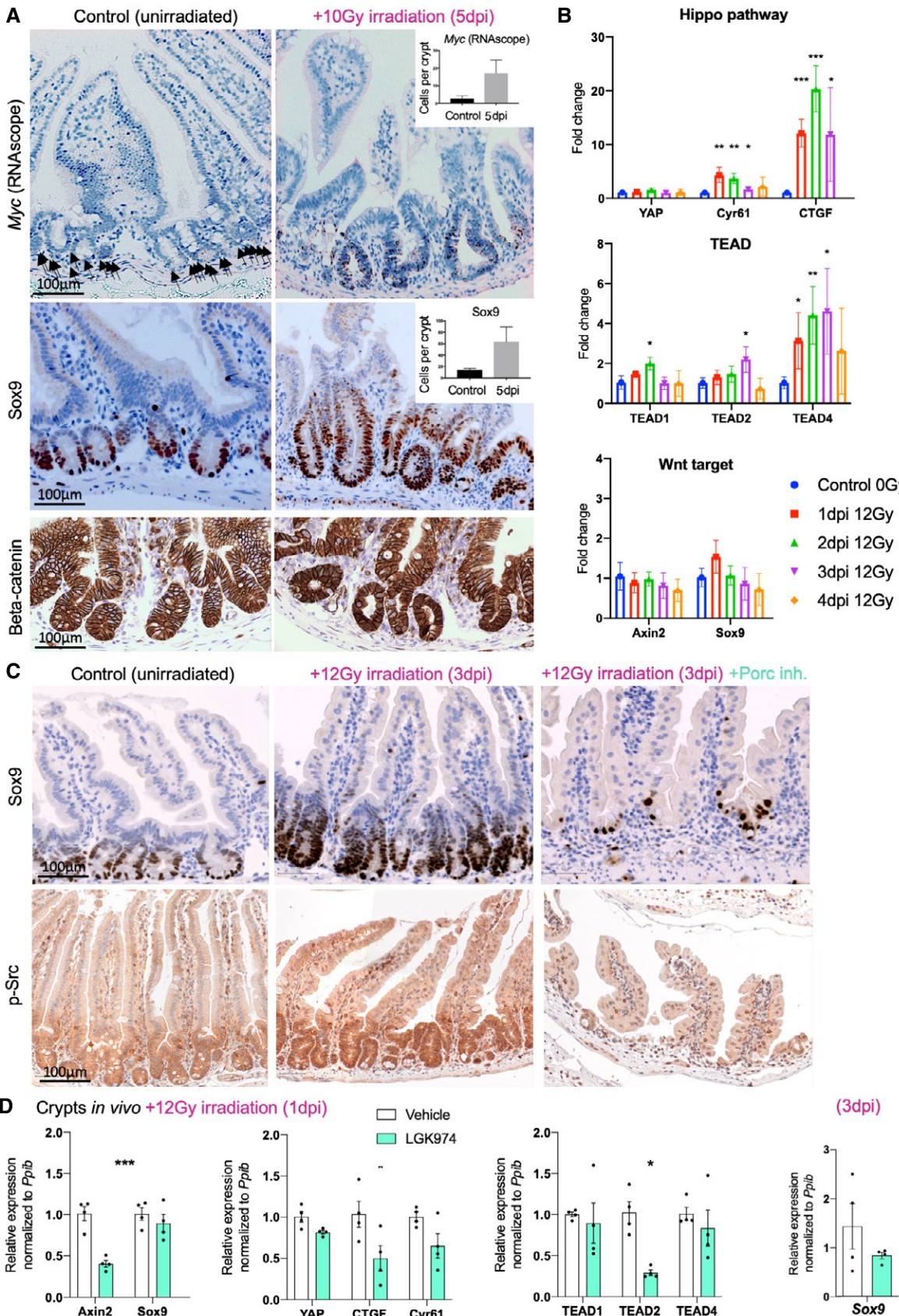

Figure 8.

**Figure 8.   Irradiation induces YAP-TEAD activity to drive crypt expansion, which also requires Wnt signalling.**

A   Control and irradiated murine small intestines analysed with RNAscope in situ hybridisation for *Myc* mRNA (black arrows point to mRNA granules in the control) and by immunostaining for Sox9 protein reveals an expansion in the expression domain of both genes. Nuclear localisation of beta-catenin is readily detectable after irradiation. $n = 3$ samples per condition, quantification shown in the insets, error bars = 1 SD. Statistical analysis was performed with an unpaired Student's *t*-test. Similar results were obtained at 3 and 5 dpi.

B   qPCR analysis of *YAP, TEADs* and their target genes *Ctgf* and *Cyr61* over a 4 days post-irradiation time course at 12 Gy. Note the strong upregulation of *TEAD4, Ctgf* and *Cyr61*. In contrast, *Axin2, TEAD2* and *Sox9* mRNA expression levels are only moderately altered, confirming that Wnt signalling remains active upon irradiation-induced YAP-TEAD activation. $n = 4$ biological replicates per time point. Statistical analysis was performed with unpaired *t*-test compared to the control 0 Gy condition, *$P < 0.05$, **$P < 0.01$, ***$P < 0.001$.

C   Control and irradiated murine small intestines analysed by immunostaining for Sox9 protein reveals that treatment with Porcupine inhibitor LGK974 strongly reduces the Sox9 expression domain induced by 3 days after 12 Gy irradiation. Phosphorylated Src immunostaining reveals an upregulation by 3 days after irradiation which is also lost upon treatment with Porcupine inhibitor LGK974 ($n = 4$ animals in each condition).

D   qPCR analysis of Wnt and YAP target gene expression in intestinal crypts isolated at 1 day after 12 Gy irradiation. Note the inhibition of *Axin2, Sox9, YAP, Ctgf, Cyr61* and *TEAD* family expression upon treatment with Porcupine inhibitor LGK974. The mild reduction of *Sox9* mRNA expression by LGK974 at 1 day after irradiation (1 dpi) becomes more pronounced at 3 days after irradiation (3 dpi). $n = 4$ biological replicates per condition, statistical analysis was performed with Student's two-tailed *t*-test, error bars = 1 SD, *$P < 0.05$, **$P < 0.01$, ***$P < 0.001$.

expansion phenotype that arises in response to YAP-TEAD activation involves an elongation/flattening of the Wnt gradient, such that transit-amplifying cells are expanded at the expense of stem cells. Synergy between Wnt and YAP signalling appears necessary for expansion of the *Sox9* and *Myc* expression domain in the regenerating crypt. Around a week later, a negative feedback process appears to act to reduce Wnt signalling and then return the crypt to its homeostatic state. In *Apc* mutant tumours, Wnt signalling is permanently active in all cells, and the progression of such tumours is known to be accelerated by chronic tissue damage responses that can activate YAP-TEAD.

# Materials and Methods

## Mouse strains

All experiments were carried out in accordance with the United Kingdom Animal Scientific Procedures Act (1986) and UK home office regulations under project license numbers 70/7926 and PDCC6E810. *Villin-Cre^ERt2^* mice were obtained from Ian Rosewell (The Francis Crick Institute). Wild-type mice were used in C57/Bl (6) background. *Src* floxed and $Fyn^{-/-}$, $Yes^{-/-}$ mice were obtained from Val Brunton (Edinburgh).

$Apc^{Min}$ mice were provided by Axel Behrens (The Francis Crick Institute) and *Lats1/2* floxed mice were provided by Randy Johnson (MD Anderson Cancer Centre).

## Inducible Cre activation

For *in vivo* experiments, Tamoxifen (Sigma, 25 mg/ml in corn oil) was injected intraperitoneally (IP; 8 μl/g body weight, or 4 μl/g for the LATS dKO experiments) for 3 or 5 consecutive days into 6- to 12-week-old controls or transgenic animals and analysed by immunohistochemistry several days thereafter. Intestinal regeneration was induced by irradiating mice with 10 to 14 Gy gamma irradiation four days after recombinase induction. Mice were sacrificed 1–5 days post-irradiation. For transgenic cultured organoids, the Cre recombination was induced by addition of 4-OHT (Sigma, H7904) at 1 μM in the culture media for 24 h after the first passage of the organoids.

## Immunohistochemistry

Control and transgenic mouse gut were harvested and cut in four sections, which were flushed with cold PBS. A metal rod was inserted into each segment and placed in a holder to cut them longitudinally to open them flat on Whatman filters. The segments were then fixed in 10% neutral-buffered formaldehyde for 24 h before being embedded in paraffin blocks. 4 μm thick sections were cut, deparrafinised and rehydrated using standard methods. After an antigen retrieval step, sections were stained with haematoxylin and eosin (H&E) solution or with primary antibody followed by a nuclear counterstaining. Additional images of human samples were obtained by data-mining the www.proteinatlas.org database.

## *In situ* hybridisation and quantification

Similar to immunohistochemistry, paraffined sections were rehydrated and subjected to the RNAscope protocol according to manufacturer's instructions. Hybridisation with PPIB (ACD, 313911) and DapB (ACD, 310043) probes as positive and negative control, respectively, was performed in parallel with the mouse *Yap1* (316601, ACD), *Tead1* (ACD, 457371), *Tead2* (ACD, 420281), *Tead4* (ACD, 312921), *Axin2* (ACD, 400331) and *Myc* (LS, 413458 2.5) probes. The brown detection kit (ACD, 322300) was used to label the targeted *Yap1* and *Myc* mRNA together with nuclear counterstaining, while the red detection kit (ACD, 322360) was used for *Tead1, Tead2, Tead4* and *Axin2*.

To quantify *YAP* mRNA expression levels, *YAP* RNAscope stained slides were scanned with the Zeiss scanner and then analysed with the StrataQuest software from TissueGnostics. 10 ROIs were manually drawn on the first segments of the gut of three mice. Within each ROI, the villi were automatically detected, while the crypts were manually drawn. In each compartment, the individual brown dots corresponding to a single mRNA molecule were detected and quantified. The quantification is the number of brown punctae per $\mu m^2$ in each compartment in every ROIs.

## Organoid experiments

Intestinal crypts were isolated from 6- to 10-week-old C57B/l6 or transgenic mice, following the published protocol from Mahe *et al*

(2013). Briefly, the whole gut was harvested and washed in cold DPBS (Gibco, 14190250). The most proximal 5 cm were cut open, and the villi were removed with a coverslip. The remaining tissue was washed and incubated in 2 mM EDTA for 30 min at 4°C. Two crypt fractions were then mechanically extracted, the first one being filtered through a 70 μM cell strainer before pooling both fractions together. After several low speed washes in ADF-12 (Gibco, 12634-010), isolated crypts were resuspended and plated in Matrigel (Corning, 354230). Organoids were cultured in IntestiCult media (Stemcell technologies, 06005) complemented with Primocin antibiotic (Invitrogen, ant-pm-05). Organoids were cultured in 24-well plates for maintaining the cultures and then cultured in 8-well chambers for drug incubations and immunostaining (Ibidi, 80827). The Porcupine inhibitor LGK974 (Selleck chemicals, S7143) was used at 5 μM for 24 h. For Src inhibition experiments, organoids were treated with 100 nM of Dasatinib (Selleck chemicals, S1021) or 100 nM eCF506 for a period of 4 h. Organoid microscopy was performed with either a Leica SP5 or a Leica SP8 laser-scanning confocal microscope.

## Antibodies

Primary antibodies used include the following: Rabbit YAP H-125 (Santa Cruz Biotechnology sc-15407), Mouse YAP 63.7 (Santa Cruz, sc-101199), Rabbit pY418 Src (Life technologies, 44660G), Mouse Ezrin (Santa Cruz, sc-58758), Rabbit Ki67 (Genetex, GTX16667) and Rabbit Cas3 (Cell signalling, 9661). Dilutions used are available from the first author upon request. For immunofluorescence secondary antibodies, Alexa-488 (Invitrogen, A32723), Alexa-568 (Invitrogen, A-11011), along with Phalloidin-647 (Invitrogen, A22287) and DAPI were used.

## RT-qPCR

Extraction of total RNA from intestinal crypts and organoids was homogenised and extracted using a RNeasy Mini Kit (Qiagen, 74106). cDNA synthesis for WT or KO mice was performed using Superscript II (Invitrogen, 18064022) or High-Capacity cDNA Reverse Transcription Kit (Applied Biosystems, 4368813). Gene samples were run in triplicates with PowerUp SYBR Green (Applied Biosystems, A25778) on a Quantstudio 12 Flex Thermocycler. Expression values and quantitation was calculated using the $\Delta\Delta CT$ method relative to the housekeeping gene (PPIB). S.E.M was used for the error bars. Yap, Tead1, Tead2, Tead3, Tead4, Cyr6, Ctgf and Lgr5 primers were purchased as QuantiTect Primers (Qiagen).

## Statistical analysis

Numerical data were plotted using Prism 9 software, and the mean and standard deviations were calculated in order to plot graphs and error bars. Student's *t*-test was performed to determine statistical significance. *$P < 0.05$, **$P < 0.01$, ***$P < 0.001$.

## Data Availability

This study does not include data deposited in external repositories.

**Expanded View** for this article is available online.

## Acknowledgements

This work was funded by the Francis Crick Institute as well as by the Cancer Research UK Beatson Institute and the Australian National University. VSWL laboratory is funded by the Francis Crick Institute, which receives its core funding from Cancer Research UK (FC001105), the UK Medical Research Council (FC001105) and the Wellcome Trust (FC001105). For the purpose of Open Access, the author has applied a CC BY public copyright licence to any Author Accepted Manuscript version arising from this submission.

## Author contributions

OG, NA, CMS, RR, AB, AK, PA and MRR performed the experiments and prepared the figures. OG, JC, VSWL and BJT conceived the experiments and wrote the manuscript with input from NA. OS, JC, VSLW and BJT obtained research funding and supervised the project.

## Conflict of interest

The authors declare that they have no conflict of interest.

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
