## [Review Process File · The EMBO Journal]

Wnt and Src signals converge on YAP-TEAD to drive intestinal regeneration

Oriane Guillermin, Nikolaos Angelis, Clara Sidor, Rachel Ridgway, Anna Baulies, Anna Kucharska, Pedro Antas, Melissa Rose, Julia Cordero, Owen Sansom, Vivian Li, and Barry Thompson

DOI: [10.15252/embj.2020105770](https://doi.org/10.15252/embj.2020105770)

Corresponding author: Barry Thompson (barry.thompson@anu.edu.au)

Review Timeline:

Submission Date:	28th May 20
Editorial Decision:	17th Jul 20
Revision Received:	18th Jan 21
Editorial Decision:	26th Feb 21
Revision Received:	11th Mar 21
Accepted:	17th Mar 21

Editor: Daniel Klimmeck

Transaction Report:

Dear Dr Thompson,

Thank you for the submission of your manuscript (EMBOJ-2020-105770) to The EMBO Journal. Please accept my sincere apologies for the extended duration of the peer-review of your study, which got delayed due to protracted reviewer input as well as detailed discussions within the editorial team. Your study has been sent to three referees for evaluation, and we have received reports from all of them, which I enclose below.

As you will see, the referees acknowledge the potential interest of your findings, although they also express major concerns. In more detail, referee #2 raises major reservations regarding the biological advance provided and details of the link between Wnt and YAP-TEAD. Referee #2 states that the depth of characterization of the VillinCre YAP-TAZ- and Lats1,2-floxed mice is too premature (ref#3, pt.1,2; see also ref#2). Further, this reviewer points to lack of mechanistic insight into the functional link between Src and nuclear YAP as well as potential upstream signals. Referee #1 states that the underlying details of how the Lats1/2-loss and Src-Wnt-dependent control of nuclear YAP are linked remains unresolved (ref#1, pt.4) and requests a more rigorous exploration on Src's function in the crypt (ref#1, pts. 3,5). In addition, the referees point to number of issues regarding technical robustness, variability of the data as well as data representation and statistics, that would need to be conclusively addressed to achieve the level of robustness and clarity needed for The EMBO Journal.

I judge the comments of the referees to be generally reasonable and given their overall interest, we are in principle happy to invite you to revise your manuscript experimentally to address the referees' comments. I need to stress though that we do need strong support from the referees on a revised version of the study in order to move on to publication of the work and as to the open outcome of the revisional work suggest to keep EMBO Reports in mind for this work as an alternative venue.

In light of the extensive experimentation requested by the reviewers, I would appreciate if you could contact me during the next weeks via e.g. a video call to discuss your perspective on the comments and potential plan for revisions.

In this context I also want to point to our adjusted GTA. We are aware that many laboratories cannot function at full efficiency during the current COVID-19/SARS-CoV-2 pandemic and have therefore extended our 'scooping protection policy' to cover the period required for a full revision to address the experimental issues highlighted in the editorial decision letter. Please contact us at any time to discuss an adapted revision plan for your manuscript should you need additional time, and also if you see a paper with related content published elsewhere.

Thank you for the opportunity to consider your work for publication. I look forward to your revision.

Kind regards,

Daniel Klimmeck

Daniel Klimmeck, PhD
Editor
The EMBO Journal

When assembling figures, please refer to our figure preparation guideline in order to ensure proper formatting and readability in print as well as on screen:
<http://bit.ly/EMBOPressFigurePreparationGuideline>

Before submitting your revision, primary datasets (and computer code, where appropriate) produced in this study need to be deposited in an appropriate public database (see <https://www.embopress.org/page/journal/14602075/authorguide#datadeposition>).

The accession numbers and database should be listed in a formal "Data Availability" section (placed after Materials & Method) that follows the model below (see also <https://www.embopress.org/page/journal/14602075/authorguide#availabilityofpublishedmaterial>). Please note that the Data Availability Section is restricted to new primary data that are part of this study.

Data availability

Our journal also encourages inclusion of *data citations in the reference list* to directly cite datasets that were re-used and obtained from public databases. Data citations in the article text are distinct from normal bibliographical citations and should directly link to the database records from which the data can be accessed. In the main text, data citations are formatted as follows: "Data ref: Smith et al, 2001" or "Data ref: NCBI Sequence Read Archive PRJNA342805, 2017". In the Reference list, data citations must be labeled with "[DATASET]". A data reference must provide the database name, accession number/identifiers and a resolvable link to the landing page from which the data can be accessed at the end of the reference. Further instructions are available at <https://www.embopress.org/page/journal/14602075/authorguide#referencesformat>

- a point-by-point response to the referees' comments, with a detailed description of the changes made (as a word file).
- a word file of the manuscript text.
- individual production quality figure files (one file per figure)
- a complete author checklist, which you can download from our author guidelines (<http://emboj.embopress.org/authorguide>).
- Expanded View files (replacing Supplementary Information)

Further information is available in our Guide For Authors:

The revision must be submitted online within 90 days; please click on the link below to submit the revision online before 15th Oct 2020.

Link Not Available

Referee #1:

In the present work, the authors show that Yap1, that is required for the intestinal regeneration, is controlled by SRC kinase via WNT activation and LATS1/LATS2 inactivation.

The manuscript is interesting, well written, most of the data are solid, and will be interesting for the general audience of EMBOJ. However, there are several points that remain unclear at this stage and the answers to these questions will be important to strengthen the manuscript and increase its

impact. These questions are detailed below.

1. The authors, on one hand, show that the increase in Yap1 expression upon APC mutations is mainly cytoplasmic. However, in the figure 4 the authors show images of APC mutated organoids and the cytoplasmic levels of Yap1 don't seem to be affected as compared to WT condition. In addition, the authors show that SRC inhibitor Dasatinib decrease nuclear Yap1 expression in APC mutant organoids. It is a bit confusing. Is APC mutation affecting nuclear or cytoplasmic Yap1?
2. The authors show that the radiation induces Yap1 nuclear localization and that this nuclear localization can be prevented by inhibition of Src. The authors also show that Src inhibitor reduces nuclear Yap1 upon Lats1/2 loss of function and upon mutation of APC. The question that arise is by which mechanism Yap1 is activated upon radiation? Is it through activation of LATS1/2? Or through activation of WNT signaling pathways? Or it's another SRC-YAP1 pathway independent of LATS1/2 or WNT?
3. Src inhibitors are drugs that are approved for clinical use in some malignancies. Following the hypothesis of the authors use of SRC-inhibitors would inhibit intestine regeneration and affect normal intestine homeostasis. Is that true? Would be interesting to see the effect of SRC inhibitors on intestine homeostasis in vivo.
4. At this stage it's also not clear how the activation of LATS1/2 and WNT signaling pathways by SRC are mechanistically linked? Does LATS1/2 loss of function alter WNT signaling? And vice versa, does APC mutation alters LATS1/2 activity?
5. The experiments with the inhibitors are very convincing. But would be interesting to see whether in the normal intestine there is a gradient of SRC activation from the crypt to the villi, as it is observed for Yap1. WB for pSRC and total SRC in the cells isolated from these 2 different compartments would be very informative.

Minor comments.

1. Most of the images across all the figures do not have scale bars. Scale bar should be added to all images.
2. Figure 3c: statistical significance should be shown, and the statistical test used should be clearly specified in the figure legend.
3. Figure 3e-g: the figure legends for these panels are missing? The panels should be clearly described in the figure legend, the test used to determine the statistical significance should be clearly indicated and the scale bar should be added to the images.
4. Figure 4a: the number of samples, as well as the exact p-value and the statistical test that has been used should be shown.
5. In the figure 4b authors show the fold change of mRNA of TEAD factors as determined by RT-PCR in WT and APC fl/fl mice and conclude that loss of APC induce strong induction. However, the standard deviation is huge, and for the Tead4 the difference is not even significantly different. Why such high variability is observed? How many samples have been analyzed? Increasing the number of samples could potentially help to reduce the variability.
6. Figure 6b and c: the p-values should be calculated and shown in the figure.

Referee #2:

Numerous reports documented that Wnt signals, YAP and Src activities form a gradient along the crypt-villus axis and how Wnt (or loss of APC) activates YAP/TAZ and how Src induces nuclear YAP via the PI3K/PDK1 axis. Hence, the observations reported in the current manuscript showing that YAP, TEAD2 and TEAD4 are transcriptionally regulated by Wnt/b-catenin signals in the intestine and nuclear translocation of YAP requires Src kinase activity are of limited novelty.

Furthermore, it is of major concern how frowsy figures and text (scale bars are missing. Figure legends are incomplete, stars and arrows in figures are not explained, etc., etc.) of this manuscript have been prepared.

Although a superficial search of the literature did not reveal report(s) that demonstrate an upregulation of TEDAs by Wnt, one would have expected that this single piece of novel data is better analyzed, e.g. by identifying Lef/Tcf binding sites in the promotor regions of TEAD genes, etc. One would have also expected that the authors correlate levels of cytoplasmic YAP with Wnt, TEAD and pSrc. This can easily be done by staining adjacent tissue sections. It is also unclear why YAP localization varies in sections derived from control tissues vary so much (Fig. 5C nuclear, Fig. 8A cytoplasmic and Figs. 6A and 7A in both compartments).

Comments:

Fig. 1:

- (1) Scale bars are missing.
- (2) Quantification of apoptosis is missing (Fig. 1A and 1B).
- (2) Since cell survival differs from previous publications, cell proliferation should also be analyzed.
- (3) Explanation for arrows in Fig. 1A and stars in Fig. 1C missing.
- (4) A high magnification image of YAP/TAZ expression of transverse fold would help interpreting the phenotype.

Fig. 2:

- (1) Nuclear YAP throughout the intestinal epithelium upon loss of LATS1/2 is well known. Nothing new in this figure.
- (2) Scale bars missing.
- (3) Quantification of Ki67 staining is missing.
- (4) Apoptosis is not analyzed.

Fig.3:

- (1) Scale bars are missing.
- (2) Legends for Fig. 3E-G are missing.
- (3) The authors refer in the main text to Fig. 3, but instead refer to Fig. 1. This looks like very careless proofreading!
- (4) The authors claim in the manuscript text that the organoids shown in Fig. 3G are of APC5 mutant origin, while the Fig. 3G indicates organoids from compound mutants lacking APC, P53, and YAP/TAZ. Why is p53 included here? And actually, it was known from previous publication that YAP is required for growth of organoids.

Fig.4

- (1) Scale bar are missing.
- (2) The result text refers to APC5 mutant organoids in Fig. 4A the organoids are in fact are from control (BRGU?) and VillinCreERT: APCflox/flox mice. The writing is terribly confusing! The
- (3) Figure legends of Fig. 4A and B are duplicated.
- (4) What does BRGU mean?
- (5) Fig.4A and B: YAP should be included into the analysis and also CTGF as positive control. What does the statistical analysis say? There are stars but it does not say what they mean?
- (6) Fig. 4C: corroboration of the conclusion that Wnt signaling induces TAED2/4 expression in vivo requires staining of adjacent sections for either Wnt ligands, Wnt effector and/or Wnt downstream targets.
- (7) Since human cancer has likely acquired mutations of tumor suppressor genes and oncogenes, staining of mouse tumors would support the conclusion.

Fig.5

- (1) Scale bars in Fig. 5A-B are missing.
- (2) Fig. 5B: expression level changed but not subcellular localization; DAPI channel and quantification of expression need to be shown.
- (3) Fig. 5C-E: culture period (days) not specified.
- (4) Fig. 5C: the authors claim that activation of Wnt signaling in APC5 organoids increases nuclear YAP. This is not convincing, as YAP is also nuclear in controls, even more prominent than that in APC5 mutants.
- (5) Fig. 5C: quantification of total and nuclear YAP is required to define ratio in both groups and to support the conclusions of the authors.

Fig.6

- (1) Fig.6B and 6C: quantifications of control or APC5? Not clear, what is actually analyzed.
- (2) Culture periods are not indicated.
- (3) Is dasatinib a Src inhibitor? To my knowledge the inhibitor was developed to inhibit Abl kinase, and later it was shown to inhibit almost the entire kinome. So, how trustful is such an inhibition?

Fig.7

- (1) Scale bars and quantifications are missing.
- (2) Wnt and pSrc staining unclear: sections or organoids?
- (3) Fig. 7B: Villin-Cre:Srcf/f without irradiation is missing. This is an important control.
- (4) Fig. 7B: Wnt and pSrc need to be analyzed for supporting the conclusions of the authors.
- (5) Fig. 7B: what are the consequences for cell proliferation, cell death and survival.
- (6) Lower and upper images are from different sections!

Fig.8

- (1) Scale bars and quantifications are missing.
- (2) Culture periods are not indicated.
- (3) Can this inhibitor also be applied in vivo? If yes, what are the consequences?

Referee #3:

In this manuscript Guillermin et al. study the regulation of the YAP pathway by Wnt signaling and Src in intestinal epithelial homeostasis, proliferation and regeneration. The authors employ conditional knockout mouse models and organoids to propose a model whereby Wnt signaling induces YAP and TEAD gene expression and primes the epithelial regenerative response whereas Src signaling promotes the nuclear translocation of YAP and intestinal proliferation. Key findings are the following:

- Ablation of both YAP and TAZ in intestinal epithelial cells leads to crypt cell death in the small intestine but not to any overt phenotype, and to spontaneous tissue damage in the proximal colon.
- Ablation of LATS1/2 in intestinal epithelial cells leads to a hyperproliferative response and to YAP nuclear localization in the epithelium.
- Wnt signaling induces the expression of TEAD2 and TEAD4 in organoids and in vivo
- Dasatinib controls YAP nuclear localization in cystic APC deficient organoids and in LATS1/2 deficient organoids.
- Ablation of Src in intestinal epithelial cells abrogates YAP nuclear localization and the regenerative response of the epithelium upon irradiation.

In vivo evidence for the role of Src as a regulator of YAP in intestinal epithelial regeneration is

missing and this is definitely a very interesting topic in the field. In that sense, the phenotypes shown by the authors are of potential interest but they need a much better characterization to be convincing and further support by alternative techniques and data quantitation. Also the study lacks in molecular mechanism: how does Src regulate YAP nuclear localization (as indicated but not directly shown by the dasatinib experiments) and what are the upstream signals? Is a defective YAP regulation at the level of nuclear translocation indeed responsible for the phenotype of VillinCre SrcFlox mice upon irradiation? A "rescue" experiment is necessary here to support this notion and advance the knowledge in the field.

Major points.

1. The authors generated VillinCreER YAPflox TAZflox mice and present data on an increased apoptotic rate at the base of the small intestinal crypts and a tissue damage phenotype in the ascending colon. These observations are interesting but the phenotypic analysis of these mice as shown in Fig.1 is poor. Also, there is no mention of the number of mice analyzed.
 - a. The data on an increased apoptotic rate should be supported by alternative techniques (TUNEL), shown in well-oriented crypts and at a magnification high enough to distinguish the epithelial specificity of the staining. The data should also be quantitated. How do the authors explain these observations? Is YAP/TAZ necessary for the survival of intestinal stem cells? If this is the hypothesis, then the authors should specifically show apoptotic stem cells by co-staining for stem cell markers and also include single knockout controls.
 - b. The phenotype of the ascending colon is described as "severe damage" and a mechanical damage hypothesis is discussed. It's hard, however, to understand the extent and the exact nature of the phenotype mentioned in the text since most of the area shown in Fig. 1C is part of the middle and distal colon, not of the ascending colon. Again, there is no mention of the number of mice examined. Was the evaluation of these specimens blinded? The specimens should be evaluated by a pathologist in a blinded fashion and quantitative data should be shown for specific histopathological criteria. This would help to exclude the possibility of having damage caused by tissue processing, especially at the gut flushing step.
2. The hyper-proliferative phenotype in the epithelium of VillinCreER Lats1flox Lats2flox mice also needs a much better characterization with alternative, quantitative techniques for proliferation markers (qPCR for Mki67 and other proliferation genes) and WB for YAP in nuclear extracts from the small intestine. Also, the area of the VillinCreER Lats1flox Lats2flox colon shown in Fig.2B (Ki67 staining) appears to be ulcerated. If this is a representative picture from the colon of these mice it is remarkable that the authors did not make any comment on this phenotype. Again, the authors should get advice from an experienced GI pathologist and explain if this ulceration is a general or a sporadic phenomenon in these mice.
3. Fig.3G: The authors conclude that YAP is essential for VillinCre APCf/f p53f/f organoids since VillinCre APCf/f p53f/f YAPf/f TAZf/f organoids do not grow. Poor growth, however, is an expected outcome for a culture of YAP deficient crypts as shown by Gregorieff and colleagues (doi:10.1038/nature15382). Although the exact conditions of organoid growth in this experiment aren't clear to me, since the legends for Fig. 3E-F are missing, it appears that the difference shown in organoid growth isn't a specific effect of YAP in the context of APC deficiency. This would be obvious if a VillinCre YAPf/f control was included in the experiment.
4. The role of Src in the nuclear localization of YAP in cystic organoids should be genetically shown in VillinCre SrcFlox organoid cultures and supported by WB data for YAP in nuclear extracts. The effect of dasatinib is not necessarily specific for Src.

We sincerely thank all three referees for their helpful comments. Please find our point-by-point responses below.

Referee #1:

In the present work, the authors show that Yap1, that is required for the intestinal regeneration, is controlled by SRC kinase via WNT activation and LATS1/LATS2 inactivation.

The manuscript is interesting, well written, most of the data are solid, and will be interesting for the general audience of EMBOJ. However, there are several points that remain unclear at this stage and the answers to these questions will be important to strengthen the manuscript and increase its impact. These questions are detailed below.

1. The authors, on one hand, show that the increase in Yap1 expression upon APC mutations is mainly cytoplasmic. However, in the figure 4 the authors show images of APC mutated organoids and the cytoplasmic levels of Yap1 don't seem to be affected as compared to WT condition. In addition, the authors show that SRC inhibitor Dasatinib decrease nuclear Yap1 expression in APC mutant organoids. It is a bit confusing. Is APC mutation affecting nuclear [or] cytoplasmic Yap1?

We thank the reviewer for this question and we are happy to clarify this point.

In a wild-type organoid, the YAP gene is more strongly expressed in the crypts compared with the 'villus' cells (Fig 5A-C). APC mutation causes increased in YAP mRNA and protein expression throughout the organoid (Fig 5C). In APC mutant organoids, this can result in higher levels of either cytoplasmic (in cuboidal cells) or nuclear localised (in stretched cells) YAP, depending on the morphology of the organoid (Fig 5E). The Fig 5D&E are from separate experiments, so the levels are not comparable. However, we now provide a quantification of the relative levels of YAP in wild-type versus APC mutant organoids result in the revised figure that confirms that the YAP levels are increased on average in this experiment, such that all cells in the organoid have high YAP expression (similar to crypt cells in a wild-type organoid) (new graph in Fig 5C).

SRC activation promotes nuclear localisation of YAP, so inhibition of SRC with Dasatinib causes reduced nuclear YAP (Fig 6). So, Wnt and Src are two pathways acting via different mechanisms (expression of YAP vs localisation of YAP) to induce YAP activity in the intestine. We have added a sentence to the discussion section to clarify this point.

2. The authors show that the radiation induces Yap1 nuclear localization and that this nuclear localization can be prevented by inhibition of Src. The authors also show that Src inhibitor reduces nuclear Yap1 upon Lats1/2 loss of function and upon mutation of APC. The question that arise is by which mechanism Yap1 is activated upon radiation? Is it through activation of LATS1/2? Or through activation of WNT signaling pathways? Or it's another SRC-YAP1 pathway independent of LATS1/S or WNT?

We thank the reviewer for raising this point. To address this question, we add new text and citations to the discussion on how irradiation leads to Yap1 nuclear localisation, including via SRC phosphorylation of LATS1/2 (Figs 6&7). We have no evidence for strongly altered Wnt signalling in this process, although we observe expanded expression of the Wnt target genes Axin2 and Sox9 upon irradiation or YAP activation - in contrast to recent claims in Cell Stem Cell from Li et al and Cheung et al that YAP completely inactivates Wnt signalling (new Fig 2A-C). Our results are instead consistent with an expansion (spreading) of the Wnt gradient, along with expansion of the entire crypt, upon YAP activation (in *Lats1/2* cKO) such that maximal Wnt levels are reduced in the crypt base while medium levels are extended along the villus – a flattening and lengthening of the gradient (Fig 2A-C).

3. Src inhibitors are drugs that are approved for clinical use in some malignancies. Following the hypothesis of the authors [use] of SRC-inhibitors would inhibit intestine regeneration and affect normal intestine homeostasis. Is that true? Would be interesting to see the effect of SRC inhibitors on intestine homeostasis in vivo.

To address this point, refer the reviewer to our genetic inhibition of Src, which is sufficient to prevent intestinal regeneration (Fig 7B). See also our previous paper on the role of c-Src in intestinal regeneration (Cordero et al 2014). We do not expect any effect on normal intestinal homeostasis with Src knockout (similar to yap/taz cKO), and our results fully align with this expectation. We were not able to obtain sufficient quantities of eCF506 to perform a large-scale in vivo analysis, owing to the COVID19-related disruption.

4. At this stage it's also not clear how the activation of LATS1/2 and WNT signaling pathways by SRC are mechanistically linked? Does LATS1/2 loss of function alter WNT signaling? And vice versa, does APC mutation alters LATS1/2 activity?

We thank the reviewer for this important question. Two very recent publications in Cell Stem Cell (Li et al and Cheung et al) indicated that YAP activation in *Lats1/2* cKO mice inhibits Wnt signalling. We provide new data that partially confirms their observations, but demands a different interpretation. We confirm that high-threshold Wnt target genes (crypt base stem cell markers) are reduced in *Lats1/2* cKO intestines, but that low-threshold Wnt target genes (Axin2, Sox9) are actually expanded (Fig 2A-C). These results suggest that YAP activation in *Lats1/2* cKO is causing an expansion of the Wnt gradient, reducing its maximal level at the crypt base but increasing its medium level along the villus (Fig 2C). Thus, our findings provide an important corrective to the conclusions of Li et al and Cheung et al that

YAP simply inhibits Wnt signalling. We have added several paragraphs to the Discussion section to compare and contrast our findings with those of Li et al and Cheung et al.

We see no evidence for APC mutations directly affecting LATS1/2 activity, as loss of APC is not sufficient to drive YAP to the nucleus in all circumstances (which would be expected if LATS1/2 are inhibited). See Fig S1C for an APC mutant colorectal adenoma with cytoplasmic YAP. YAP only becomes nuclear following either tissue damage/inflammation or progression to invasive CRC.

5. The experiments with the inhibitors are very convincing. But would be interesting to see whether in the normal intestine there is a gradient of SRC activation from the crypt to the villi, as it is observed for Yap1. WB for pSRC and total SRC in the cells isolated from these 2 different compartments would be very informative.

We thank the reviewer for this question. To address this point, we provide new data (Fig 8C) to show the active phosphorylated SRC (p-SRC) gradient in both normal small intestine and during intestinal regeneration. We also add new discussion to suggest a possible explanation for the gradient of p-SRC, whose activation in response to stimulus can be promoted by the Wnt target gene CD44.

Minor comments.

1. Most of the images across all the figures do not have scale bars. Scale bar should be added to all images.

Done.

2. Figure 3c: statistical significance should be shown, and the statistical test used should be clearly specified in the figure legend.

Done. Note this figure is now Fig S6.

3. Figure 3e-g: the figure legends for these panels are missing? The panels should be clearly described in the figure legend, the test used to determine the statistical significance should be clearly indicated and the scale bar should be added to the images. Done. Note this figure is now Fig S6.

4. Figure 4a: the number of samples, as well as the exact p-value and the statistical test that has been used should be shown.

Done.

5. In the figure 4b authors show the fold change of mRNA of TEAD factors as determined by RT-PCR in WT and APC fl/fl mice and conclude that loss of APC induce strong induction. However, the standard deviation is huge, and for the Tead4 the difference is not even significantly different. Why such high variability is observed? How many samples have been analyzed? Increasing the number of samples could potentially help to reduce the variability.

Done. We have added extensive new data in figures 3 and 4 on regulation of TEAD mRNA.

6. Figure 6b and c: the p-values should be calculated and shown in the figure.

Done.

Referee #2:

Numerous reports documented that Wnt signals, YAP and Src activities form a gradient along the crypt-villus axis and how Wnt (or loss of APC) activates YAP/TAZ and how Src induces nuclear YAP via the PI3K/PDK1 axis. Hence, the observations reported in the current manuscript showing that YAP, TEAD2 and TEAD4 are transcriptionally regulated by Wnt/b-catenin signals in the intestine and nuclear translocation of YAP requires Src kinase activity are of limited novelty.

We agree with the reviewer that there have been numerous reports of Wnt, YAP and Src action in the intestine, but there remains great confusion and controversy over the role and regulation of these pathways. For example, two very recent papers in Cell Stem Cell (Li et al and Cheung et al) claim that YAP inhibits Wnt signalling and that YAP therefore acts as a tumour suppressor in intestinal organoids and organoid-derived tumours (Cheung et al). These findings conflict with previous papers showing that YAP has an oncogenic (pro-proliferative) role in the intestinal regeneration and in APC mutant tumour formation. To help resolve these discrepancies, we provide new data to show that the Wnt gradient is actually flattened and expanded upon YAP activation (presumably owing to more rapid cell proliferation and crypt expansion) (new Fig 2C,D). In addition, we provide new data to show that YAP does not act as a tumour suppressor in intestinal organoids and organoid-derived tumours – in contrast to Cheung et al (new Fig S8, S9 & S10).

Furthermore, it was previously completely unclear how Wnt and Src are integrated to regulate YAP – and this is the main point of novelty in our manuscript, which (as the reviewer states) focuses on regulation of YAP, TEAD2 and TEAD4 by Wnt/b-catenin and nuclear localisation of YAP by Src. Although the reviewer considered the novelty limited, we feel that these are in fact very important mechanisms to resolve, and the reviewer themselves points out just a sentence later that these are indeed novel findings:

Although a superficial search of the literature did not reveal report(s) that demonstrate an upregulation of [TEADs] by Wnt, one would have expected that this single piece of novel data is better analyzed, e.g. by identifying Lef/Tcf binding sites in the promotor regions of TEAD genes, etc.

We thank the reviewer for raising this point. In the revised manuscript, we provide extensive new data on the regulation of TEAD mRNAs by Wnt and by YAP itself. We agree with the reviewer that our study of the upregulation of

TEAD2 and TEAD4 by Wnt could benefit from identification of LEF/TCF sites in the promoter regions, but this would require significant additional ChIP experiments which are outside the scope of the current manuscript.

One would have also expected that the authors correlate levels of cytoplasmic YAP with Wnt, TEAD and pSrc. This can easily be done by staining adjacent tissue sections.

To address this point, we provide extensive new analysis of the expression profiles of YAP, beta-catenin, TEAD4 and pSrc along the crypt-villus axis in both resting and regenerative conditions, as well as in APC mutant conditions.

It is also unclear why YAP localization varies in sections derived from control tissues vary so much (Fig. 5C nuclear, Fig. 8A cytoplasmic and Figs. 6A and 7A in both compartments).

We agree with the reviewer that there is variability in the YAP subcellular localisation within control organoids, which we have made explicitly clear in the first panel introducing them (Fig 5A), where we show there is a time-course over which YAP is initially cytoplasmic but then becomes nuclear in organoids cultured for more than 3 days in Matrigel. In the old Fig 8A, the control is cytoplasmic because it has just been passaged into new Matrigel. We have removed this panel from the revised manuscript to avoid any potential confusion. The model is that organoids secrete their own ECM that then stimulates Integrin signalling to drive YAP to the nucleus (see work from Kim Jensen's lab). Notably, in the Lats1/2 cKO organoids there is no variability: all the YAP is constitutively nuclear regardless of the number of days in culture (Fig 6D).

Comments:

Fig. 1:

(1) Scale bars are missing.

Fixed.

(2) Quantification of apoptosis is missing (Fig. 1A and 1B).

Quantification of apoptosis is now provided in the revised Fig 1A and B.

(2) Since cell survival differs from previous publications, cell proliferation should also be analysed.

Ki67 staining is now provided in the revised Fig 1A and B.

(3) Explanation for arrows in Fig. 1A and stars in Fig. 1C missing.

Fixed.

(4) A high magnification image of YAP/TAZ expression of transverse fold would help interpreting the phenotype.

See revised Fig 1D.

Fig. 2:

(1) Nuclear YAP throughout the intestinal epithelium upon loss of LATS1/2 is well known. Nothing new in this figure.

We now provide more novel data in this figure, particularly data that addresses the two very recent papers in Cell Stem Cell (Li et al and Cheung et al) which claim that activation of YAP in Lats1/2 cKO mice inhibits Wnt signalling in the intestine. We show that, although stem cell markers are reduced, the classical Wnt target genes Axin2 and Sox9 are actually expanded along the crypt-villus axis, which is consistent with a flattening and extension of the Wnt signalling gradient along this axis (new Fig 2A-C).

(2) Scale bars missing.

Fixed.

(3) Quantification of Ki67 staining is missing.

We consistently see a very strong increase in Ki67 positive cells in crypts. Quantified in the new Fig S4B.

(4) Apoptosis is not analyzed.

We now provide the cleaved caspase 3 (Cas3) immunostaining to analyse apoptotic cells in the revised Fig 2A.

Fig.3:

(1) Scale bars are missing.

Fixed.

(2) Legends for Fig. 3E-G are missing.

Fixed.

(3) The authors refer in the main text to Fig. 3, but instead refer to Fig. 1. This looks like very careless proofreading!

We apologise for this inadvertent labelling error, which crept in during versioning of the manuscript. This is now fixed.

(4) The authors claim in the manuscript text that the organoids shown in Fig. 3G are of APC5 mutant origin, while the Fig. 3G indicates organoids from compound mutants lacking APC, P53, and YAP/TAZ. Why is p53 included here? And actually, it was known from previous publication that YAP is required for growth of organoids.

We apologise for this mistake in presentation. The figure labelling is correct and this has been fixed in the text. APC, p53 mutant tumour-derived organoids are analysed because both APC and p53 are frequently mutated in colorectal cancer, and we wished to establish whether YAP was required for tumour-derived organoid growth, even when both these tumour suppressors are mutated. This result therefore has novelty compared with published data, and is also important as it conflicts with two very recent papers in Cell Stem Cell claiming that activation of YAP inhibits Wnt signalling (Li et al and Cheung et al) such that YAP is a tumour suppressor that inhibits organoid growth (Cheung et al). We see the identical effect in the absence of p53, such that yap/taz are required for Apc knockout organoid growth.

Fig.4

(1) Scale bar are missing.

Fixed.

(2) The result text refers to APC5 mutant organoids in Fig. 4A the organoids are in fact are from control (BRGU?) and VillinCreERT: APC^{flox/flox} mice. The writing is terribly confusing!

We apologise for the confusion. We have now revised the text to clarify that the organoids in Fig 4A are APC5 while the *in vivo* mouse intestines in Fig 4B are from APC^{flox/flox} mice. We have also amended the labelling of the Fig 4A and B to clarify this point.

(3) Figure legends of Fig. 4A and B are duplicated.

This mistake has now been corrected.

(4) What does BRGU mean? It is a wild-type strain of mice from the Biological Resources General Use (BRGU) stock. We have removed any reference to this internal-use acronym from the revised version of the manuscript.

(5) Fig.4A and B: YAP should be included into the analysis and also CTGF as positive control. What does the statistical analysis say? There are stars but it does not say what they mean?

We now add extensive new analysis of YAP and TEAD mRNA expression, with CTGF as a positive control.

(6) Fig. 4C: corroboration of the conclusion that Wnt signaling induces [TEAD2/4] expression *in vivo* requires staining of adjacent sections for either Wnt ligands, Wnt effector and/or Wnt downstream targets.

To address this point, we provide images of immunostaining of the Wnt downstream target Sox9 in a similar set of sections to those of TEAD4 (revised Fig 4C).

(7) Since human cancer has likely acquired mutations of tumor suppressor genes and oncogenes, staining of mouse tumors would support the conclusion.

We provide extensive new TEAD mRNA *in situ* and Q-PCR analysis in mouse *Apc* mutant tumours, as requested (Fig 4A,B).

Fig.5

(1) Scale bars in Fig. 5A-B are missing.

Fixed.

(2) Fig. 5B: expression level changed but not subcellular localization; DAPI channel and quantification of expression need to be shown.

To address this point, we provide quantification of the expression level of YAP, as requested (revised Fig 5B). There was not sufficient space to incorporate the DAPI channel in this panel; however, a DAPI channel is shown the next panel (Fig 5C).

(3) Fig. 5C-E: culture period (days) not specified.

Fixed in revised Fig 5C. D and E are examples of different organoid ages, to observe different morphologies.

(4) Fig. 5C: the authors claim that activation of Wnt signaling in APC5 organoids increases nuclear YAP. This is not convincing, as YAP is also nuclear in controls, even more prominent than that in APC5 mutants.

To clarify, we do not claim that Wnt signalling increases nuclear YAP in these organoids. In fact, the opposite is true that YAP is nuclear in both controls and APC5 mutant organoids – as the reviewer has pointed out. We have amended the relevant portion of the results section to make this clearer. Instead, it is mechanical forces and other non-Wnt inputs that are of primary importance in governing YAP nuclear localisation in APC5 mutant organoids.

(5) Fig. 5C: quantification of total and nuclear YAP is required to define ratio in both groups and to support the conclusions of the authors. See answer to point (4) above.

Fig.6

(1) Fig.6B and 6C: quantifications of control or APC5? Not clear, what is actually analyzed.

To clarify, these are the control crypt buds that are analysed. This was stated in the figure legend but we have now added this information to the figure panels to make it clearer.

(2) Culture periods are not indicated.

Fixed.

(3) Is dasatinib a Src inhibitor? To my knowledge the inhibitor was developed to inhibit Abl kinase, and later it was shown to inhibit almost the entire kinome. So, how trustful is such an inhibition?

Yes, Dasatinib is a potent Src family kinase inhibitor, that also inhibits Abl tyrosine kinase. Our experiments with this drug are supported by conditional Villin-Cre Src^{flox/flox} knockout experiments, which confirm that Src activity is required for YAP nuclear localisation in the intestinal epithelium (Fig 7B). In addition, we now include treatment with a much more specific Src inhibitor, eCF506, which has a similar effect on YAP as Dasatinib (Fig 6C).

Fig.7

(1) Scale bars and quantifications are missing.

Scale bars are now provided in the revised Fig 7A&B. Quantification of the effect of Dasatinib on YAP is now shown in the new Fig 6B.

(2) Wnt and pSrc staining unclear: sections or organoids?

In these sections and organoids we stain for YAP and F-actin/DAPI. We now provide extensive new analysis of Wnt signalling (beta-cat staining) and Wnt target genes *Myc* and *Sox9*, as well as p-Src, in normal and regenerating intestines in the revised Fig 8A-C.

(3) Fig. 7B: Villin-Cre:Src^f without irradiation is missing. This is an important control.

To address this point, we cite the previous manuscript on Src (Cordero et al) showing this control.

(4) Fig. 7B: Wnt and pSrc need to be analyzed for supporting the conclusions of the authors.

We now provide extensive new analysis of Wnt signalling (beta-cat staining) and Wnt target genes *Myc* and *Sox9*, as well as p-Src, in normal and regenerating intestines in the revised Fig 8A-C.

(5) Fig. 7B: what are the consequences for cell proliferation, cell death and survival.

To address this point, we cite the previous manuscript on Src (Cordero et al) showing this analysis.

(6) Lower and upper images are from different sections!

Yes, this is done on purpose to show an additional example for each genotype/treatment. This is now explicitly pointed out in the figure legend for clarity.

Fig.8

(1) Scale bars and quantifications are missing.

Fixed.

(2) Culture periods are not indicated.

Fixed.

(3) Can this inhibitor also be applied in vivo? If yes, what are the consequences?

Dasatinib is too toxic for in vivo use, but we have employed another more specific Src family kinase inhibitor eCF506 that does not cause toxicity in vivo. The eCF506 inhibitor works well in vivo but is the subject of a separate manuscript.

Referee #3:

In this manuscript Guillermin et al. study the regulation of the YAP pathway by Wnt signaling and Src in intestinal epithelial homeostasis, proliferation and regeneration. The authors employ conditional knockout mouse models and organoids to propose a model whereby Wnt signaling induces YAP and TEAD gene expression and primes the epithelial regenerative response whereas Src signaling promotes the nuclear translocation of YAP and intestinal proliferation. Key findings are the following:

- Ablation of both YAP and TAZ in intestinal epithelial cells leads to crypt cell death in the small intestine but not to any overt phenotype, and to spontaneous tissue damage in the proximal colon.
- Ablation of LATS1/2 in intestinal epithelial cells leads to a hyperproliferative response and to YAP nuclear localization in the epithelium.
- Wnt signaling induces the expression of TEAD2 and TEAD4 in organoids and in vivo
- Dasatinib controls YAP nuclear localization in cystic APC deficient organoids and in LATS1/2 deficient organoids.
- Ablation of Src in intestinal epithelial cells abrogates YAP nuclear localization and the regenerative response of the epithelium upon irradiation.

In vivo evidence for the role of Src as a regulator of YAP in intestinal epithelial regeneration is missing and this is definitely a very interesting topic in the field. In that sense, the phenotypes shown by the authors are of potential interest but they need a much better characterization to be convincing and further support by **alternative techniques and data quantitation**.

We thank the reviewer for this point. We now provide this in our revised manuscript.

Also the study lacks in molecular mechanism: how does Src regulate YAP nuclear localization (as indicated but not directly shown by the dasatinib experiments) and what are the upstream signals? **Is a defective YAP regulation at the level of nuclear translocation indeed responsible for the phenotype of VillinCre SrcFlox mice upon irradiation? A "rescue" experiment is necessary here to support this notion and advance the knowledge in the field.**

To address the reviewer's point, we provide new data in the revised manuscript. In addition to our data showing that YAP/TAZ is required for Src-dependent regeneration (such that yap/taz dKO has the same phenotype as SrcFlox mice), we now provide an in vivo mouse genetic experiment involving expression of active YAP (nlsYki5SA) to show that this transgene is sufficient to mimic Src activation during regeneration (Fig S5). Thus YAP is necessary and sufficient for Src signalling to control intestinal regeneration. Details on Src regulation of YAP localisation are outside the scope of this manuscript, but recent publications show that it functions by directly phosphorylating and inactivating LATS1/2 kinases – a mechanism that our own data support, as the effect of a specific Src inhibitor on YAP nuclear localisation is abolished upon lats1/2 dKO (Fig 6C).

Major points.

1. The authors generated VillinCreER YAPflox TAZflox mice and present data on an increased apoptotic rate at the base of the small intestinal crypts and a tissue damage phenotype in the ascending colon. These observations are interesting but the phenotypic analysis of these mice as shown in Fig.1 is poor. Also, there is no mention of the number of mice analyzed.

We thank the reviewer for pointing this out. We have now significantly updated the revised Fig 1, including quantification of apoptosis, and additional data is also supplied Fig S2 and S3. The figure legends provide the N numbers, as requested.

a. The data on an increased apoptotic rate should be supported by alternative techniques (TUNEL), shown in well-oriented crypts and at a magnification high enough to distinguish the epithelial specificity of the staining. The data should also be quantitated.

How do the authors explain these observations? Is YAP/TAZ necessary for the survival of intestinal stem cells? If this is the hypothesis, then the authors should specifically show apoptotic stem cells by co-staining for stem cell markers and also include single knockout controls.

We have used anti-cleaved Caspase 3 immunostaining to identify apoptotic cells, which is a standard method in the field, so it is a little unclear to us why TUNEL is also necessary. We now provide quantification of the number of apoptotic cells per crypt in the revised Fig 1A&B.

b. The phenotype of the ascending colon is described as "severe damage" and a mechanical damage hypothesis is discussed. It's hard, however, to understand the extent and the exact nature of the phenotype mentioned in the text since most of the area shown in Fig. 1C is part of the middle and distal colon, not of the ascending colon. Again, there is no mention of the number of mice examined. Was the evaluation of these specimens blinded? The specimens should be evaluated by a pathologist in a blinded fashion and quantitative data should be shown for specific histopathological criteria. This would help to exclude the possibility of having damage caused by tissue processing, especially at the gut flushing step.

We thank the reviewer for raising this point. We have updated the Fig 1 legend to provide the N numbers, as requested. We now provide quantification of the frequency of abnormal ascending colon phenotypes in the mutants, as judged by our pathologist, in the revised manuscript results section. Additional examples of the colon phenotype of the VillinCreER YAPflox TAZflox mice after tissue damage are provided in the new Fig S3. We did not see any effect of tissue processing or gut flushing on any of our wild-type samples, suggesting that this is not an issue with our analysis.

2. The hyper-proliferative phenotype in the epithelium of VillinCreER Lats1flox Lats2flox mice also needs a much better characterization with alternative, quantitative techniques for proliferation markers (qPCR for Mki67 and other proliferation genes) and WB for YAP in nuclear extracts from the small intestine.

Also, the area of the VillinCreER Lats1flox Lats2flox colon shown in Fig.2B (Ki67 staining) appears to be ulcerated. If this is a representative picture from the colon of these mice it is remarkable that the authors did not make any comment on this phenotype. Again, the authors should get advice from an experienced GI pathologist and explain if this ulceration is a general or a sporadic phenomenon in these mice.

We again thank the reviewer for pointing this out. To address this point, we provide a quantification of the number of Ki67+ cells in control versus VillinCreER Lats1flox Lats2flox mice (new Fig S4B). We agree with the reviewer that the VillinCreER Lats1flox Lats2flox mice do display intestinal inflammation (causing diarrhea) and occasional ulceration, which we provide a clear example of this phenomenon in sections immunostained for YAP and Ki67 in the new Fig S4B.

3. Fig.3G: The authors conclude that YAP is essential for VillinCre APCf/f p53f/f organoids since VillinCre APCf/f p53f/f YAPf/f TAZf/f organoids do not grow. Poor growth, however, is an expected outcome for a culture of YAP deficient crypts as shown by Gregorieff and colleagues (doi:10.1038/nature15382). Although the exact conditions of organoid growth in this experiment aren't clear to me, since the legends for Fig. 3E-F are missing, it appears that the difference shown in organoid growth isn't a specific effect of YAP in the context of APC deficiency. This would be obvious if a VillinCre YAPf/f control was included in the experiment.

We thank the reviewer for pointing this out. We do cite the very important Gregorieff and Co paper frequently in the revised manuscript, which indeed shows a key role for YAP in organoid growth. We have now corrected the figure legend. Unfortunately, space restrictions prevent us from showing all these controls in this figure. However, we do show both VillinCreER+ controls and YAPf/f TAZf/f controls for our in vivo experiments in Fig 1.

4. The role of Src in the nuclear localization of YAP in cystic organoids should be genetically shown in VillinCre SrcFlox organoid cultures and supported by WB data for YAP in nuclear extracts. The effect of dasatinib is not necessarily specific for Src.

We thank the reviewer for this suggestion. To address this point, we have treated organoids with a new and highly specific Src inhibitor (eCF506) that confirms our observations with Dasatinib (Fig 6C&D). Unfortunately, we could not grow VillinCre Src flox organoids to analyse the nuclear localisation of YAP; in other words, the VillinCre SrcFlox organoid cultures grow very poorly, similar to the VillinCre YAPflox TAZflox organoids, as expected from our model that Src acts upstream of YAP/TAZ to drive intestinal cell proliferation during regeneration or in organoids.

Dear Dr Thompson,

Thank you for submitting your revised manuscript (EMBOJ-2020-105770R) to The EMBO Journal. Please accept my apologies for getting back to you with unusual protraction due to delayed reviewer input during re-review as well as detailed discussions here in the team. Your amended study was sent back to the three referees, and we have received comments from two of them, which I enclose below. While the third referee was unfortunately not able to provide feedback at this time, we have carefully evaluated your response to his-her critique and found the issues raised to be satisfactorily addressed.

As you will see the other referees stated that the manuscript has been significantly improved, and they are now in favour of publication, pending satisfactory minor revision.

Thus, we are pleased to inform you that your manuscript has been accepted in principle for publication in The EMBO Journal.

Please consider the remaining points of the referee #2 by complementing annotation of statistics, methods and data display.

Related, we need you to take care of a number of points related to formatting and data representation as detailed below, which should be addressed at re-submission.

Further, I will share additional changes and comments from our production team during the next days to be considered.

Please contact me at any time if you have additional questions related to below points.

Thank you for giving us the chance to consider your manuscript for The EMBO Journal. I look forward to your adjusted manuscript files.

Again, we are happy to swiftly move forward with acceptance of this work upon re-submission. Please contact me at any time if you need any help or have further questions.

Kind regards,

Daniel Klimmeck

>> Please specify author contributions for all authors.

>> Specify up to five keywords for the article.

>> Recheck callouts and their correct order in the main text for Fig. 4D and individual EV figure panels.

>> Please enter the complete funding information in the Acknowledgements section of your manuscript.

>> There are 12 EV figures. Please limit to maximally five EV figures and compile the remaining ones in an Appendix .pdf file with ToC on its first page. Remove the legends removed from the manuscript and add them to the appendix. Figure callouts need to be corrected accordingly to "Figure EV1"-5 and "Appendix Figure S1"-7.

>> Please clarify re-display of 3d YAP/actin-stained control group organoids from Figure 6A in the figure legend for Figure 7A.

>> Please introduce a separate 'Data accessibility' section in the Material and Methods part, stating 'This study does not include data deposited in external repositories.'

>> Provide a 'Statistical Analysis' section detailing the algorithms applied.

>> Please remove the statement 'Expression of nlsYAP5SA was confirmed by immunostaining (not shown):' from p.29, or add respective data to Figure S9.

>> Add a 'Conflict of Interest' paragraph to your manuscript.

- a point-by-point response to the referees' comments, with a detailed description of the changes made (as a word file).

- a word file of the manuscript text.

- individual production quality figure files (one file per figure)

- a complete author checklist, which you can download from our author guidelines (<https://www.embopress.org/page/journal/14602075/authorguide>).

- Expanded View files (replacing Supplementary Information)

The revision must be submitted online within 90 days; please click on the link below to submit the revision online before 26th May 2021.

Link Not Available

Referee #1:

In the revised version of the manuscript Guillermin O et al include additional results that significantly strengthen the data. They were not able to perform all the requested experiments due to the COVID restrictions, but they provide additional discussion on the raised questions based on their previous experience and published data. All the minor comments raised have been addressed. I recommend publication of this paper in EMBO Journal.

Referee #2:

The experiments demonstrate that Wnt and Src cooperate to facilitate YAP activity in epithelial cells of the stressed intestine. Although the manuscript improved and it was a bit easier to read it remains in an unacceptably sloppy condition: the number of experiments in most figure legends are missing, gene sequences for RT-PCR, antibody information, and probe information for Axin and Myc are missing, the cartoon and Fig. 4D mentioned in the results and discussion, respectively, are missing while figures such as Fig. S11 and S12 are not mentioned in the manuscript, references are wrong, rectangles in Fig. 5C are not explained, etc. etc. Most concerning, however, is the fact that the control image in Fig. 6A and 7A is a duplication. The image is flipped and strangely has a different size in 6A and 7B. This fully damages the confidence in this manuscript.

The authors performed the requested editorial changes.

Dear Dr Thompson,

Thank you for submitting the revised version of your manuscript. I have now evaluated your amended manuscript and concluded that the remaining minor concerns have been sufficiently addressed.

Thus, I am pleased to inform you that your manuscript has been accepted for publication in the EMBO Journal.

Please note that it is EMBO Journal policy for the transcript of the editorial process (containing referee reports and your response letter) to be published as an online supplement to each paper.

Also in case you might NOT want the transparent process file published at all, you will also need to inform us via email immediately. More information is available here:

http://emboj.embopress.org/about#Transparent_Process

Please note that in order to be able to start the production process, our publisher will need and contact you regarding the following forms:

- PAGE CHARGE AUTHORISATION (For Articles and Resources)

[http://onlinelibrary.wiley.com/journal/10.1002/\(ISSN\)1460-2075/homepage/tej_apc.pdf](http://onlinelibrary.wiley.com/journal/10.1002/(ISSN)1460-2075/homepage/tej_apc.pdf)

- LICENCE TO PUBLISH (for non-Open Access)

Your article cannot be published until the publisher has received the appropriate signed license agreement. Once your article has been received by Wiley for production you will receive an email from Wiley's Author Services system, which will ask you to log in and will present them with the appropriate license for completion.

- LICENCE TO PUBLISH for OPEN ACCESS papers

Authors of accepted peer-reviewed original research articles may choose to pay a fee in order for their published article to be made freely accessible to all online immediately upon publication. The EMBO Open fee is fixed at \$5,200 (+ VAT where applicable).

We offer two licenses for Open Access papers, CC-BY and CC-BY-NC-ND.

For more information on these licenses, please visit: <http://creativecommons.org/licenses/by/3.0/> and http://creativecommons.org/licenses/by-nc-nd/3.0/deed.en_US

- PAYMENT FOR OPEN ACCESS papers

You also need to complete our payment system for Open Access articles. Please follow this link and select EMBO Journal from the drop down list and then complete the payment process:

https://authorservices.wiley.com/bauthor/onlineopen_order.asp

Notably, please be reminded that under the DEAL agreement of European scientific institutions with our publisher Wiley, you could be eligible for free publication of your article in the open access format. Please contact either the administration at your institution or Wiley (embojournal@wiley.com) to clarify further questions.

Should you be planning a Press Release on your article, please get in contact with embojournal@wiley.com as early as possible, in order to coordinate publication and release dates.

On a different note, I would like to alert you that EMBO Press is currently developing a new format for a video-synopsis of work published with us, which essentially is a short, author-generated film explaining the core findings in hand drawings, and, as we believe, can be very useful to increase visibility of the work. This has proven to offer a nice opportunity for exposure i.p. for the first author of the study. Please see the following link for representative examples:
https://www.embopress.org/video_synopses

If you have any questions, please do not hesitate to call or email the Editorial Office.

Kind regards,

Daniel Klimmeck

Daniel Klimmeck, PhD
Senior Editor
The EMBO Journal
EMBO
Postfach 1022-40
Meyerhofstrasse 1
D-69117 Heidelberg
contact@embojournal.org
Submit at: <http://emboj.msubmit.net>

Corresponding Author Name: Barry J Thompson

Manuscript Number: EMBOJ-2020-105770